# An intact S-layer is advantageous to *Clostridioides difficile* within the host

**Michael J. Ormsby**[1☯¤a], **Filipa Vaz**[1☯¤b], **Joseph A. Kirk**[2], **Anna Barwinska-Sendra**[3], **Jennifer C. Hallam**[1], **Paola Lanzoni-Mangutchi**[3¤c], **John Cole**[1], **Roy R. Chaudhuri**[2], **Paula S. Salgado**[3], **Robert P. Fagan**[2], **Gillian R Douce**[1]*

**1** School of Infection and Immunity, College of Medical, Veterinary & Life Sciences, University of Glasgow, Scotland, United Kingdom, **2** Molecular Microbiology, School of Biosciences, University of Sheffield, England, United Kingdom, **3** Biosciences Institute, Faculty of Medical Sciences, Newcastle University, England, United Kingdom

☯ These authors contributed equally to this work.
¤a Current address: Current addresses: Biological and Environmental Sciences, Faculty of Natural Sciences, University of Stirling, Stirling, United Kingdom
¤b Current address: Department of Immunology, University of Oslo and Oslo University Hospital, Oslo, Norway
¤c Current address: Université Grenoble Alpes, CNRS, CEA, IBS, Grenoble, France
* E-mail: Gillian.Douce@glasgow.ac.uk

**Data Availability Statement:** With respect to data availability statement, please amend to X-ray structural data has been deposited in the PDB repository under PDB ID 8BBY. Both the Raw RNA

## Abstract

*Clostridioides difficile* is responsible for substantial morbidity and mortality in antibiotically-treated, hospitalised, elderly patients, in which toxin production correlates with diarrhoeal disease. While the function of these toxins has been studied in detail, the contribution of other factors, including the paracrystalline surface layer (S-layer), to disease is less well understood. Here, we highlight the essentiality of the S-layer *in vivo* by reporting the recovery of S-layer variants, following infection with the S-layer-null strain, FM2.5. These variants carry either correction of the original point mutation, or sequence modifications which restored the reading frame, and translation of *slpA*. Selection of these variant clones was rapid *in vivo*, and independent of toxin production, with up to 90% of the recovered *C. difficile* population encoding modified *slpA* sequence within 24 h post infection.

Two variants, subsequently named FM2.5varA and FM2.5varB, were selected for study in greater detail. Structural determination of SlpA from FM2.5varB indicated an alteration in the orientation of protein domains, resulting in a reorganisation of the lattice assembly, and changes in interacting interfaces, which might alter function. Interestingly, variant FM2.5varB displayed an attenuated, FM2.5-like phenotype *in vivo* compared to FM2.5varA, which caused disease severity more comparable to that of R20291. Comparative RNA sequencing (RNA-Seq) analysis of *in vitro* grown isolates revealed large changes in gene expression between R20291 and FM2.5. Downregulation of *tcdA*/*tcdB* and several genes associated with sporulation and cell wall integrity may account for the reported attenuated phenotype of FM2.5 *in vivo*. RNA-seq data correlated well with disease severity with the more virulent variant, FM2.5varA, showing s similar profile of gene expression to R20291 *in vitro*, while the attenuated FM2.5varB showed downregulation of many of the same virulence associated traits as FM2.5. Cumulatively, these data add to a growing body of evidence that the S-layer contributes to *C. difficile* pathogenesis and disease severity.

seq and whole genomic sequencing data have been deposited within Gene Expression Omnibus (GEO) repository and can be accessed using the reference ID GSE205747. https://www.ncbi.nlm.nih.gov/geo/query/acc.cgi?acc=GSE205747. All other relevant data are available within the manuscript and its Supporting information files.

**Funding:** This work was supported by the Wellcome Trust, UK [204877/Z/16/Z] awarded to GRD, PSS, RPF. This grant funded research posts for FV., MJO., ABS., JAK., JH. The funders had no role in study design, data collection and analysis, decision to publish or preparation of the manuscript.

**Competing interests:** The authors have declared that no competing interests exist.

## Author summary

The S-layer of *C. difficile* is a paracrystalline array that covers the outer surface of the bacterial cell but its contribution to overall disease remains unclear. As previously described, spontaneous *slpA*-null mutant, FM2.5, with a point mutation in *slpA* offered an opportunity to study the role of the S-layer in disease. Here, we confirm that this strain is less virulent *in vivo* despite effectively colonising the host and producing toxin. We also show *in vivo* selection for sequence modifications that restore *slpA* translation and produce an S-layer. While such modifications do not affect the overall 3D structure of individual SlpA (sub)domains, they can lead to altered orientation of the structural domains and subsequent S-layer assembly. Importantly, RNA-Seq analysis *in vitro* showed large differences in gene expression between FM2.5 and R20291. Detected differences in transcription of genes involved in toxin expression and sporulation suggests that the S-layer provides a selective survival advantage within the host, which contributes to disease severity.

## Introduction

*Clostridioides difficile* is the most common cause of hospital acquired diarrhoea globally, with disease linked to disruption of the intestinal microbiota through antibiotic use [1]. The virulence of *C. difficile* has widely been attributed to the production of two toxins; toxins A (TcdA, enterotoxin) and B (TcdB, cytotoxin), responsible for cytoskeletal modifications, epithelial damage, inflammation, and fluid loss [2, 3]. A third toxin, the binary *C. difficile* toxin (CDT), expressed by only a subset of strains, has been linked to enhanced disease severity [3]. Consequently, *C. difficile* colitis has widely been considered as a toxin-mediated disease. However, the availability of tools to analyse gene expression and improved methods of mutagenesis [4], together with the availability of an accessible murine animal model [5, 6], have offered new opportunities to identify other traits, both bacterial and host-associated, that impact disease severity [7, 8]. The recent use of such approaches has provided clearer understanding of the metabolic flexibility of these organisms, the role of the microbiome in disease progression [8–10] and established several bacterial factors that influence the host response [11–13]. Of particular interest in this context, is the role of the S-layer in disease. This paracrystalline protein array is the outermost layer of the *C. difficile* cell envelope, with similar structures found in many bacteria and virtually all archaea [14].

The S-layer has been shown to perform multiple and vital roles including providing protection from environmental factors such as variations in pH, mechanical and osmotic stresses [15–17]. *In vivo*, it is proposed to play a role in molecular sieving [18] and ion trapping, protecting the organism from antimicrobial peptides and bacteriolytic enzymes produced in response to infection [19, 20]. The S-layer has also been shown to be a key target in bacteriophage predation [20–22].

In *C. difficile*, the main component of the S-layer is SlpA, which is post-translationally cleaved by a cell wall protein (CWP), Cwp84, into two functional S-layer proteins (SLPs), $SLP_L$ and $SLP_H$ [14, 23]. The proteinaceous array is further decorated by other CWPs, which provide additional functionality [14]. Assembly of the paracrystalline array relies on tiling of $SLP_H$ triangular prisms on the cell wall, interlocked by $SLP_L$ ridges facing the environment [24]. Exposure of $SLP_L$ to the environment is consistent with its high sequence variability observed between different *C. difficile* strains, with 13 different S-layer cassette types (SLCTs) identified

to date [20, 25]. Strikingly, the lattice is very compact compared to other studied S-layers, which have pores of between 30–100 Å compared with only ~10 Å in diameter in *C. difficile* [24]. This tight packing correlates well with the hypothesis that S-layer acts as a molecular sieve [18], as deletion of the most exposed regions of $SLP_L$ results in a strain with increased sensitive to lysozyme, in comparison to the parent strain, R20291 [24].

In *C. difficile*, the S-layer has also been implicated in host cell adhesion [26], biofilm formation [23, 27, 28] and immunomodulation through cell signalling of the host response [29–31]. SlpA has been shown to induce innate and adaptive immune responses through activation of TLR4 [32]. However, the role of the S-layer in *C. difficile* pathogenesis and in immune evasion remains poorly understood.

Previously, we reported the isolation and characterization of a spontaneous *C. difficile* strain lacking an S-layer, FM2.5 [20]. In studies using the Golden Syrian hamster as the infection model, FM2.5 caused no symptoms of disease, despite effectively colonising infected animals [20]. However, the acute sensitivity of hamsters to *C. difficile* toxins and lack of readily available immunological tools limits their usefulness in studying the more nuanced facets of this infection. In contrast, mice are naturally less susceptible to CDI, requiring more extensive antibiotic treatment to suppress the flora, and higher challenge doses to achieve colonisation [6, 33]. However, mice offer greater opportunities to determine the contributions of other virulence-associated traits on disease outcome, including long-term persistence associated with relapsing disease [34].

Here, we sought to elucidate the role of the S-layer as a major virulence determinant in a murine model of infection, to determine whether the loss of virulence observed in the hamster model is reciprocal in other hosts. Our results suggest that the S-layer offers a competitive colonisation advantage within the mouse intestine and is important for *in vivo* disease severity.

## Results

### The S-layer contributes to severe disease in the murine model of *C. difficile*

In a murine model of infection, loss of body weight offers a strong correlative measure of *C. difficile* disease severity [6, 33, 35]. Infection of antibiotic pre-treated mice with strain R20291 resulted in significant weight loss of up to 15%, peaking between 24 and 48 h post-infection (hpi). In contrast, mice infected with the S-layer deficient derivative strain FM2.5 showed consistently and significantly less weight loss than R20291, peaking at around 6% at 48 hpi, which is not statistically significant different to that determined in the PBS control group (Fig 1a). It is worth noting that, despite this modest weight loss, animals infected with FM2.5 failed to return to their pre-infection weight, even after the R20291-infected mice had fully recovered (96 hpi, Fig 1a). Although measurement of total *C. difficile* in faecal material showed comparable levels of total numbers of bacteria shedding at 24 and 72 hpi (Fig 1b), the recovered number of spores was significantly lower in animals infected with FM2.5 at 24 h. This suggests that while FM2.5 appears to be growing in the lumen, it is not sporulating as efficiently as R20291; as previously reported [20]. In addition, a significant reduction in both total and spore counts were also observed at 24 hpi in both caecal (Fig 1c) and colonic (S1 Fig) luminal contents from mice infected with FM2.5 compared to R20291, indicating that this strain was less prevalent. In contrast, by 48 hpi, comparable numbers of both FM2.5 and R20291 (total and spore counts) were recovered from both luminal contents. Equivalent recoveries were also observed in faecal material assessed at 72 hpi and in luminal contents at 96 hpi. It should be noted that acute disease, typified by diarrhoea at 48 hpi, made comparative analysis of faecal material unfeasible at this time point.

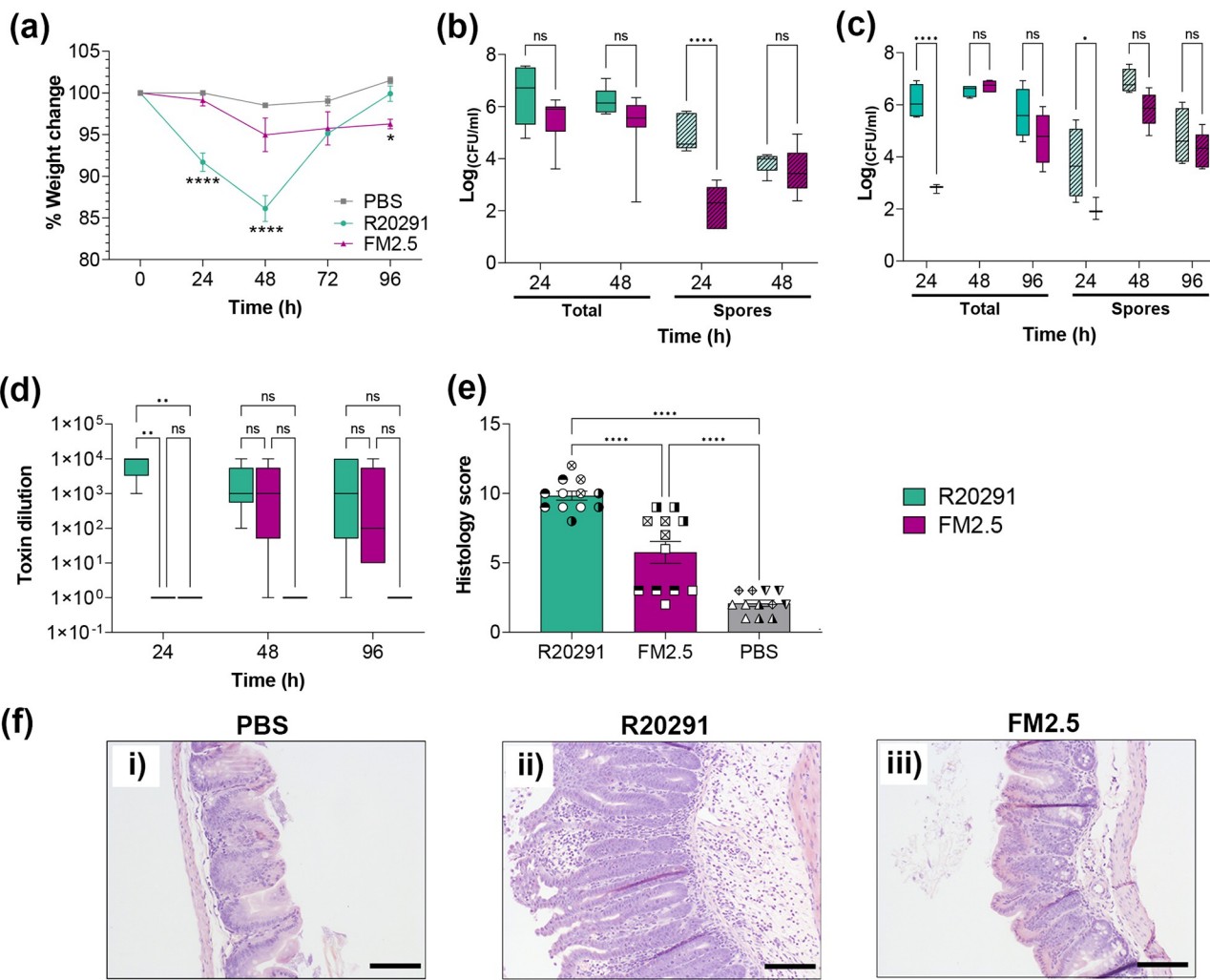

**Fig 1. SlpA deficient *C. difficile* FM2.5 causes less severe disease in a murine model of infection.** Female C57/Bl6 mice were challenged with spores of R20291 (green) or FM2.5 (purple), or mock infected with sterile PBS (grey). (**a**) Weight loss was monitored every 24 h for four consecutive days following infection. Each point is the average of several replicate experiments (n>3), with at least 10 animals per time point. Infection with R20291 strain results in statistically significant weight loss at 24 and 48 hpi, compared to both FM2.5 and PBS, while no significant difference was observed between FM2.5 and PBS at these time points. Mice infected with FM2.5 showed a slightly, but statistically significant, lower weight at 96 hpi, compared to the PBS control group. (**b**) CFU ml⁻¹ of total (no pattern) and spore recovery (pattern) from faecal material collected at 24 hpi (n = 8, R20291; n = 15, FM2.5) and 72 hpi (n = 7, R20291; n = 14, FM2.5). Variation in sample numbers reflects either difficulties associated with collection from infected mice due to diarrheal symptoms, or loss of individual animals due to disease severity. (**c**) CFU ml⁻¹ of total (no pattern) and spore recovery (pattern) in caecal contents at 24, 48 and 96 hpi (n = 5 at each time point except R20291 at 24 hpi; n = 4). (**d**) Toxin activity of caecal content at 24, 48 and 96 hpi; through challenge of Vero cells *in vitro* (n = 5 at each time point). Results displayed indicate the reciprocal of lowest dilution at which toxin activity could be measured. (**e**) Histological scoring of sections of the caecum from mice challenged with PBS, R20291 or FM2.5. Results displayed are the mean ± SEM of sections from at four mice per group (indicated by different symbols) with three sections scored from each tissue. (**f**) Histopathological sections representing caecal (i, ii and iii) sections following challenge with PBS (i); R20291 (ii); or FM2.5 (iii). Scale bars represent 100 μm. Statistical tests were conducted using GraphPad Prism software v.12. Statistical significance is indicated: ns—not significant; *p < 0.05; **p < 0.01; and ***p < 0.001.

Interestingly, mice infected with R20291 that survived infection showed full recovery by 96 hpi, returning to pre-infection weights, equivalent to those of non-infected mice. In contrast, FM2.5-infected mice failed to return to their pre-infection weight even when animals were monitored for a further five days (9 days pi) despite animals remaining asymptomatic, with no evidence of loose faeces.

Assessment of *in vivo* toxin production showed that less toxin was detected in mice infected with FM2.5 in both the caecum (Fig 1d) and in the colon (S1b Fig) at 24 hpi compared to

R20291. However, at 48 and 96 hpi, comparable levels of toxin were recovered from all infected animals. Histological examination at 48 hpi of caecal tissue from R20291 infected mice showed greater tissue damage and inflammation than those infected with FM2.5 or control mice challenged with PBS. Cumulative scoring of caecal and colonic tissue histology are shown in Fig 1e and S1c Fig, indicating changes are significant in caecal and lower colon tissue, and show a non-significant trend in the upper colon. Typical images of tissue damage are also displayed in Fig 1f (caecum) and S1d Fig (lower colon).

### *In vivo* pressure drives selection for S-layer variants

When grown on selective chromogenic agar (ChromID; BioMerieux), the morphology of R20291 presents with the typical 'fried egg' *C. difficile* colony, visible after approximately 16 h of incubation (Fig 2a). In contrast, FM2.5 produces smaller and smoother colonies, which take ~24 h to emerge (Fig 2b). Following infection of mice with R20291 and FM2.5, faecal material was recovered and plated daily. During examination of resultant colonies, it was noted that, while colonies from R20291 infected mice showed the expected morphology, material retrieved from FM2.5-infected mice showed a mixture of both large (FM2.5$_{large}$) and small colony types (FM2.5$_{small}$) (Fig 2b). Additionally, FM2.5$_{large}$, were countable after 16 h incubation, while the expected FM2.5-like colonies, FM2.5$_{small}$, were only observable from 24 h. Several colonies from both types, FM2.5$_{small}$ and FM2.5$_{large}$, were streaked from the original plates and sub-cultured twice to ensure clonality. Colony morphology of individual clones remained consistent on subsequent subcultures, suggesting this was not a phase variable trait. Individual clones were then stored at -80°C.

Colonies of different sizes were observed in several independent animal experiments, from which several clones were recovered. Amplification of the *slpA* sequence from these clones revealed that the FM2.5$_{large}$ colonies contained modifications in the genomic sequence upstream of the FM2.5 mutation site (single nucleotide insertion, Fig 2c). The first variant identified FM2.5*slpA*246delT (subsequently referred to as FM2.5$_{varA}$) showed a single nucleotide deletion (246delT), which restored the original reading frame and rescued translation of the full SlpA; modifying three amino acid residues in the translated protein. An insertion of five-nucleotides was observed in a second variant, FM2.5*slpA*249insCTTAG (subsequently referred to as FM2.5$_{varB}$; 249-253insCTTAG), resulting in the modification of 13 amino acids within the mature protein (Fig 2c). Genomic sequencing of individuals clones of these variants confirmed that they were closely related to FM2.5 and were not unrelated strains carried as contaminants by the mice (S2 Fig).

To confirm restoration of SlpA expression in these strains, low pH cell surface extracts of FM2.5$_{varA}$ and FM2.5$_{varB}$ were analysed by SDS-PAGE. This confirmed that both SLP$_H$ and SLP$_L$ proteins were present in R20291, absent in FM2.5 but restored in FM2.5$_{varA}$ and FM2.5$_{varB}$ (Fig 2d). The proteins were confirmed as SLP$_H$ and SLP$_L$ by western immunoblot analysis using anti-SLP$_H$ and anti-SLP$_L$ antibodies (Fig 2e and 2f).

Interestingly, these and other variants were also identified in subsequent *in vivo* experiments, even when animals were infected with batches of independently prepared spores. This reproducible recovery of S-layer variants *in vivo* raised the possibility that low numbers of genetic variants, present within the FM2.5 population, were being amplified within the *in vivo* environment. To test this hypothesis, we undertook amplicon sequencing of *slpA* in a preparation of FM2.5 spores used to infect mice. and from faecal material recovered from these animals at 24, 48, 72 and 96 hpi (Fig 2g,). Analysis of this unbiased amplification of *slpA* amplicons from faeces from these mice, revealed a third variant, FM2.5*slpA*252delA (subsequently named FM2.5$_{varC}$), in which the original frameshift mutation was corrected by

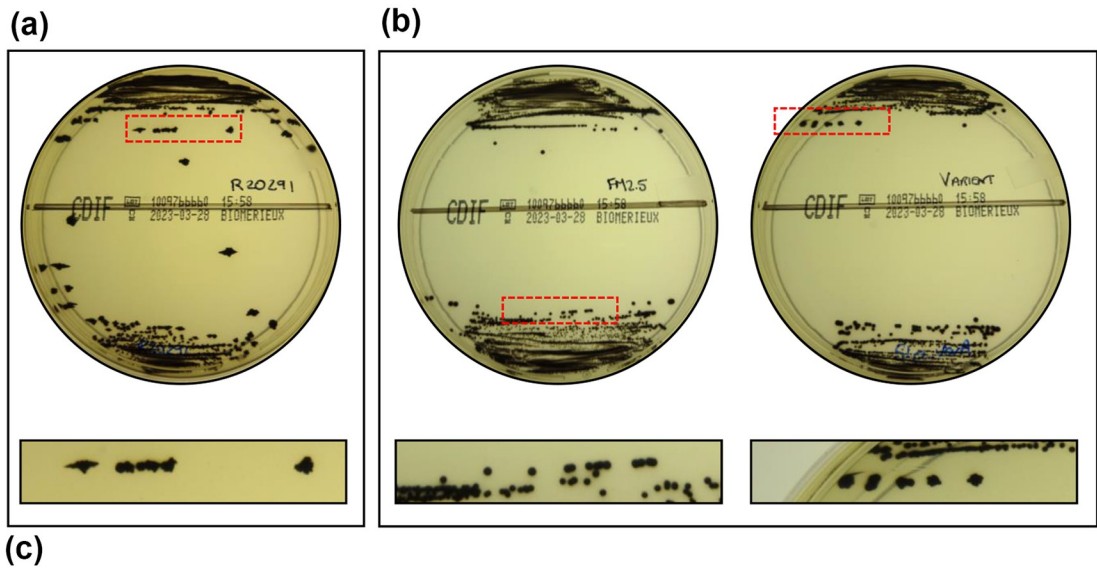

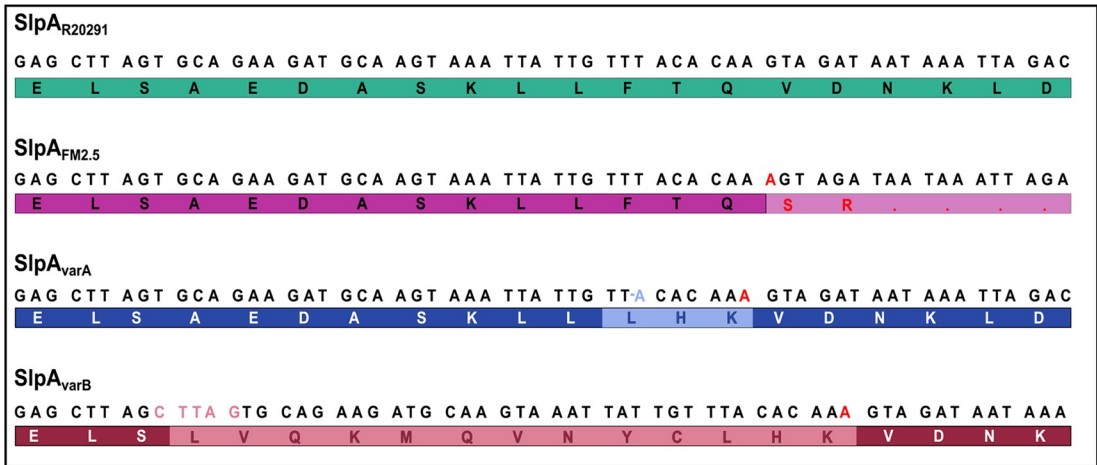

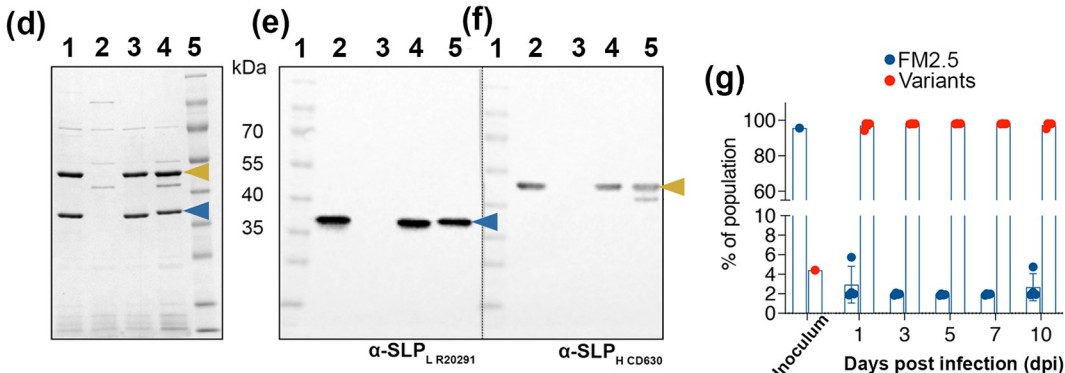

**Fig 2. Recovery of S-layer variants following *in vivo* challenge.** Following challenge with spores of R20291 and FM2.5, faecal material was recovered and plated on *C. difficile* selective chromogenic agar (Biomerieux). (**a**) Colonies of R20291. A dashed red box indicates the area enlarged to show colony morphology in the panel below. (**b**) The two colony types of FM2.5. An equivalently sized dashed red boxed area has been enlarged and colonies from each plate are shown below. FM2.5 is indicative of the typical FM2.5-like, smaller colony morphology, while the FM2.5$_{variant}$ is representative of the larger colony type. (**c**) Sequencing of a region of *slpA* shows the nucleotide and protein sequence of SlpA$_{R20291}$ (green); the nucleotide insertion (red) in SlpA$_{FM2.5}$ (light purple), introducing a stop codon that results in premature translational termination, leading to truncation of the SlpA protein. A single nucleotide deletion (light blue) in the SlpA$_{FM2.5}$ sequence, results in the modification of 3 amino acids

(shown in light blue) in SlpA$_{varA}$, (dark blue); a five-nucleotide insertion (light crimson) in SlpA$_{FM2.5}$ sequence results in modification of 13 amino acids (shown in light crimson) in SlpA$_{varB}$ (crimson). These modifications in the sequence of SlpA$_{varA}$ and SlpA$_{varB}$, restore the reading frame of *slpA*, allowing transcription and translation of the entire protein. (**d**) SDS-PAGE analysis of surface layer proteins extracted by low pH preparation. Lane 1: MW Marker; Lane 2: R20291; Lane 3: FM2.5; Lane 4: FM2.5$_{varA}$; Lane 5: FM2.5$_{varB}$. (**e**) Western immunoblot analysis using an anti-SLP$_H$ antibody. (**f**) Western immunoblot analysis using an anti-SLP$_L$ antibody. Bands corresponding to SLP$_L$ and SLP$_H$ indicated with blue and gold arrowheads, respectively. (g) Relative proportion of SlpA$_{FM2.5}$ and SlpA$_{varC}$ sequences in samples analysed in both the spore preparations used for mouse inoculations, and in faecal samples recovered up to 10 days post infection. Raw images are provided in S3 Fig.

deletion of the extra nucleotide (252delA). This variant was also identified, albeit at a low proportion of the population ($< 5\%$, Fig 2g), in the spores used for inoculation of these animals. Neither of the previous variants FM2.5$_{varA}$ or FM2.5$_{varB}$ were detected in either the spore or faecal samples tested, although this does not exclude the possibility that these or others were present in this preparation but that detection was beyond the discretionary power of the sequencing data. Based on our overall observations, each spore preparation is likely to contain low numbers of different variants, clones of which outgrow and outcompete the mutant within the population *in vivo*. However, isolation of variants as early as 24 hpi ($> 94\%$ of the population) suggests that expression of an intact S-layer provides a competitive advantage *in vivo* for these clones which express an intact S-layer.

## Sequence modifications can affect SlpA structure and assembly

To understand the effects of the detected variations in amino acid sequence on SlpA structure and S-layer assembly, crystallisation of SlpA$_{R20291}$, SlpA$_{varA}$ and SlpA$_{varB}$ was carried out. Although crystals were obtained for all strains, only SlpA$_{varB}$ crystals were of sufficient quality for X-ray diffraction data collection and structural determination by molecular replacement, using previous SlpA structures as models. As we have so far been unable to obtain an experimental structure of the complete SlpA$_{R20291}$, our previously described variant—SlpA$_{R\Delta D2}$—(PDB ID: 7ACZ), which lacks the most exposed region of SLP$_L$ [24], was used in our comparisons as a model of the S-layer protein in R20291. SLP$_H$ and the interacting domains were easily traceable in the electron density but D1 was only partially built, whilst density for domain D2 was very poor and this region could not be traced in the final SlpA$_{varB}$ model (Fig 3a, PDB ID: 8BBY and S1 Table). This implies that D2 is flexible and/or unstructured, while the structure of the core domains required for S-layer assembly—SLP$_H$, and, to a lesser extent, D1 and LID/HID [24]—seems to be generally maintained. However, the relative orientation of these domains in the SlpA$_{varB}$ molecule is altered (Fig 3a), with D1 and the interacting domains rotated towards the SLP$_H$ plane by ~30 ˚ (Fig 3a). The 13 altered residues in α2$_L$ in SlpA$_{varB}$ result in disruption of the α-helix secondary structure and introduce disorder in the upstream loop that links the preceding β-strand (β3$_L$) and α2$_L$. It is worth noting that SLP$_H$ in R20291, which belongs to SLCT 4, has several insertions within the cell wall binding 2 (CWB2) sequence motifs that define CWPs in *C. difficile*, when compared to other SlpA types. These insertions could not be traced in our previous SlpA$_{R\Delta D2}$ model [24] but were traceable in SlpA$_{varB}$ and result in several loops protruding above the SLP$_H$ plane, towards the environment, partially occluding the CWB2 motifs (Fig 3a, right). Together with the movement of the interacting domains and D1 towards the SLP$_H$ tiles, this creates a more compressed arrangement (~66 Å thickness compared to ~76 Å in SlpA$_{CD630}$, PDB ID: 7ACY, Fig 3b, bottom).

In the previously described crystallographic models of SlpA$_{CD630}$ (PDB ID: 7ACY), SlpA$_{R7404}$ (PDB ID: 7ACX) and SlpA$_{R\Delta D2}$ (PDB ID: 7ACZ), α2$_L$ was responsible for closing a

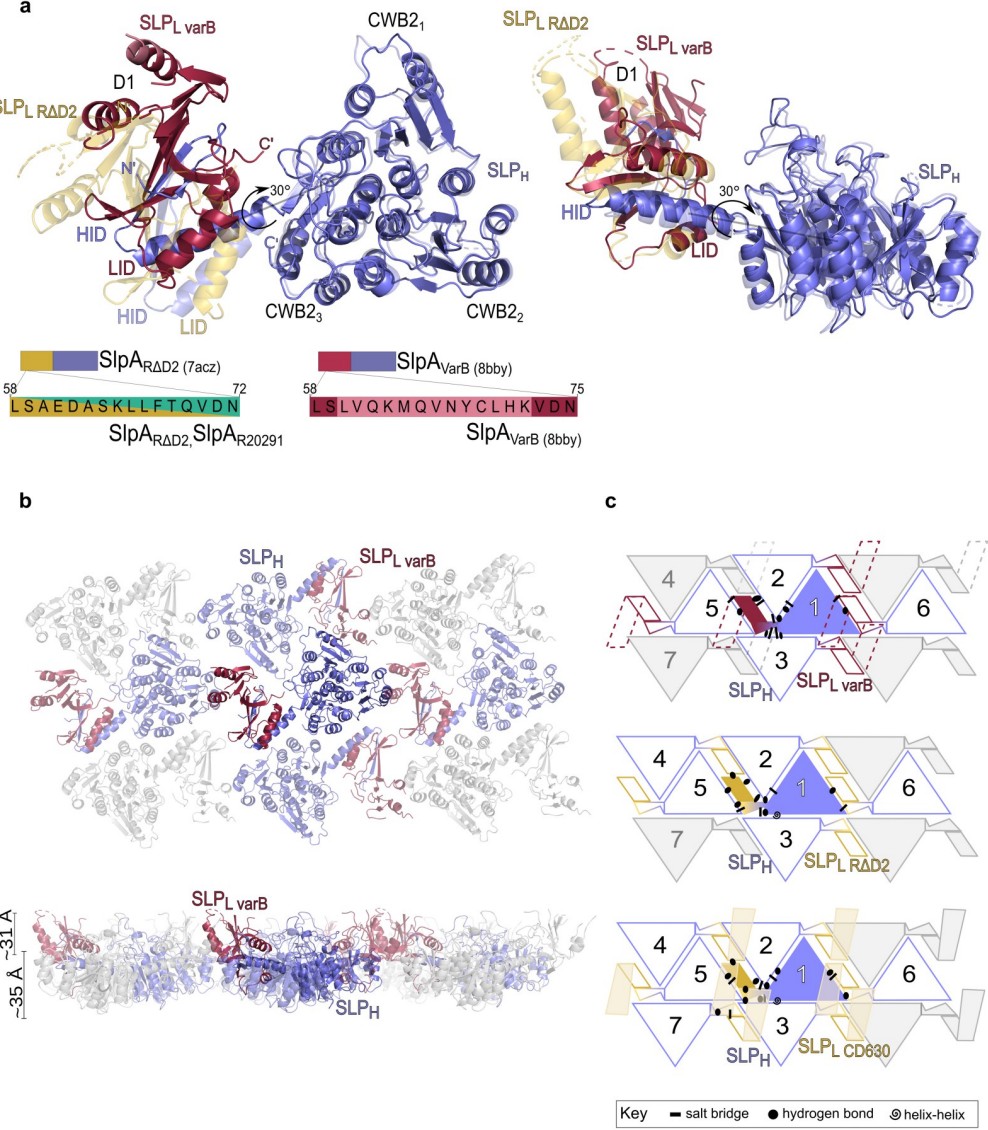

**Fig 3. Structure of SlpA$_{varB}$ shows a different assembly arrangement.** (**a**) Structural model of SlpA$_{varB}$ (SLP$_L$—pale red, SLP$_H$—slate blue, PDB ID: 8BBY), superimposed on the R20291-derived SlpA$_{RΔD2}$ model (SLP$_L$—gold, SLP$_H$—slate blue, semi-transparent), with the rotation angle of the D1 and LID/HID domains shown by an arrow. Three distinct structural features are observed: SLP$_H$, LID/HID and D1. Cartoon representation of the SLP$_H$/SLP$_L$ (H/L) complex, as seen from the environmental side (left) and side view (right). Sequence of α2$_L$, with paler colours indicating differences, is shown schematically. (**b**) Cartoon representation of the H/L planar array (PDB ID 8BBY, interacting molecules coloured and viewed as in **a**). (**c**) 2D schematic of H/L complex crystal packing in SlpA$_{varB}$ (top), SlpA$_{RΔD2}$ (centre) and SlpA$_{CD630}$ (bottom), indicating the interaction network linking a single H/L (slate blue/crimson or slate blue/gold) complex with neighbouring molecules in a planar arrangement generated by SLP$_H$ tiling. The missing D2 in the SlpA$_{varB}$ model is represented as dashed lines. Notably, D1-D1 interactions seen in other models are missing in SlpA$_{varB}$ and the SLP$_H$ tiles are shifted, with new HID-CWB2$_3$ interactions stabilising the lattice. Array is depicted as seen from the extracellular environment, with symbols representing key interaction types in the crystal lattice, detailed in S2 Table.

gap between neighbouring SlpA molecules via D1-D1 interactions [24]. In the crystallographic model of the R20291-derived SlpA$_{RΔD2}$ variant, D1-D1 interactions are mediated by hydrogen bonds between S50$_L$-S50$_L$ and Q70$_L$-A49$_L$ from neighbouring molecules.

In the SlpA$_{varB}$ structure, disruption of α2$_L$ and reorientation of D1 and LID/HID relative to SLP$_H$ leads to changes in the interactions between neighbouring molecules and,

consequently, a rearrangement of the S-layer array. Strikingly, the D1-D1 interactions seen in previous models were not observed here, possibly due to the flexibility of $\alpha2_L$ and preceding loop caused by the changes in the variant sequence leading to a different orientation of D1 domains. Unlike in the previously determined structures, neighbouring D1 domains in the $SlpA_{varB}$ structure are too far apart to mediate contacts ($> 12$ Å). In the previous models, $SLP_H$ tiling creates two wide channels, which are stabilised by interactions mediated by the interacting domains and D1 [24]. A different mode of stabilising the $SLP_H$ tiling is observed in $SlpA_{varB}$, with the interacting domains now partially inserted in those cavities (Fig 3b and S4a Fig). A new interacting interface between HID from one molecule and $CWB2_3$ from a neighbouring molecule occludes these gaps and stabilises the S-layer lattice (Fig 3c and S4a Fig). This new arrangement of the crystal lattice is in line with our proposed assembly model, where the S-layer 2D array is maintained mostly by hydrogen bonds and salt bridges across surfaces with complementary charges [24], largely dependent on $SLP_H$-$SLP_H$ interactions and stabilised by varying degree of interactions involving $SLP_L$ (Fig 3c and S2 Table). The structural model of $SlpA_{varB}$ confirms that changes in $SLP_L$ can be accommodated with minor structural changes to the (sub)domains, by exploring flexible loops and hinges to provide a stable S-layer.

As no crystal data was obtainable, we initially used AlphaFold 2 [36] to predict models for $SlpA_{R2021}$ and $SlpA_{varA}$. However, in the absence of a specific template, the predicted models show considerable variations of the relative orientation of $SLP_L$ and $SLP_H$ or some of the proteins sub-domains. The observed orientations are not compatible with our knowledge that $SLP_H$ faces the cell wall and $SLP_L$ the environment [24, 37]. This misorientation was also observed in AlphaFold 2 predictions of our experimental structures of $SlpA_{CD630}$ and the $SlpA_{varB}$ and therefore we did not proceed with AlphaFold 2 predicted models. Instead, we used the SWISS-MODEL server, either without a template input, or using $SlpA_{R\Delta D2}$ [24] or the $SlpA_{varB}$ structure determined here. Depending on which template was used (automatically selected $SlpA_{R7404}$, PDB ID: 7ACX; $SlpA_{R\Delta D2}$ or $SlpA_{varB}$), different predicted structures of $SlpA_{varA}$ were obtained, varying mostly in the orientation of D1 and interacting domains relative to $SLP_H$ (S4b Fig). Interestingly, one common feature was that the changes resulting from the altered sequence seem to be accommodated not by altering the α-helix but by varying the length of the upstream loop that links the preceding β-strand ($\beta3_L$) and $\alpha2_L$ (S4b Fig). It is therefore unclear if $SlpA_{varA}$ is more likely to adopt a R20291-like as observed in the $SlpA_{R\Delta D2}$ model or $SlpA_{varB}$-like S-layer assembly, as both can accommodate the modified sequence.

### *In vivo* S-layer selection is independent of toxin expression

FM2.5 has previously been reported to show a delay in toxin production [20], consequently we chose to investigate whether recovery of an intact S-layer was a toxin-dependent, or toxin-independent event. To answer this question, mice were infected with FM2.5Δ*PaLoc*, in which the **Pa**thogenicity **Loc**us (PaLoc), encoding toxins A and B, had been deleted. These animals, in contrast to those infected with FM2.5, showed no weight loss over the 96 hours of infection (Fig 4a) and as expected, no toxin was observed in samples from the caecum or colon (S5a Fig). Total numbers of bacteria (S5b Fig) and spores (S5c Fig) recovered from the caecum and colon of these mice were comparable at 96 hpi to that observed in animals infected with FM2.5.

Interestingly, S-layer variants were also recovered from these mice, with sequence modifications in the same region of *slpA* as previously identified. This variant FM2.5Δ*Paloc slpA*237-delTTAT (subsequently referred to as FM2.5Δ*Paloc*$_{varD}$) had four nucleotide deletions (Fig 4b; *A*237delTTAT). These changes also restore an intact SlpA, indicating that any potential selection advantage is independent of toxin production.

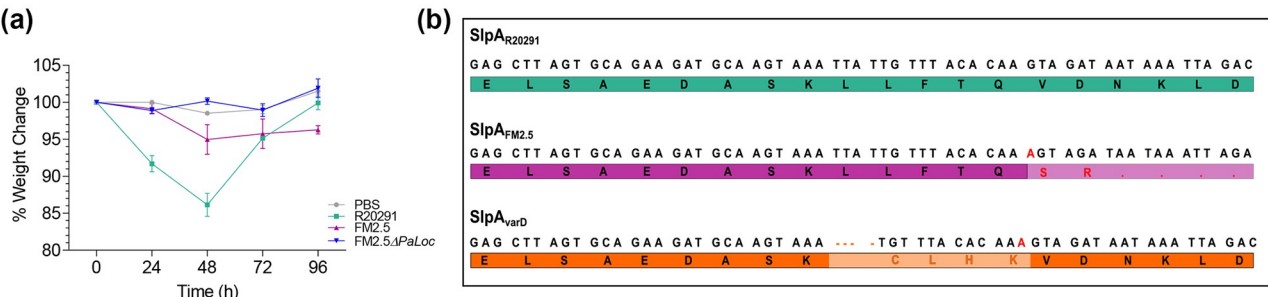

**Fig 4. *In vivo* challenge of mice with FM2.5ΔPaLoc.** Female C57/Bl6 mice were challenged with spores of R20291 (green), FM2.5 (purple) and FM2.5*ΔPaLoc* (blue) or PBS (grey). (**a**) Weight loss was monitored every 24 h for four consecutive days following infection. Each point is the average change of weight, calculated from a minimum of 5 mice per group. (**b**) Comparison of the sequences of *slpA* shows the sequence for R20291 (green) and the reported insertion in FM2.5 (red) responsible for the truncation of the SlpA protein; deletion of four nucleotides (orange) leads to changes in four amino acids (light orange) in the sequence in FM2.5ΔPaloc_varD (orange). For comparison, the intact sequence of *slp*A from R20291 is shown in green. Statistical tests were conducted using GraphPad Prism software v.12. Statistical significance is indicated: ns—not significant; *p < 0.05; **p < 0.01; and ***p < 0.001.

## The variant strains display differing levels of virulence in mice

To assess whether recovery of SlpA by these variants correlated with rescued virulence, mice were infected with spore preparations of FM2.5_varA and FM2.5_varB, alongside R20291 and FM2.5 (Fig 5a). Interestingly, infection with FM2.5_varA resulted in significant weight loss within the first 48 h of infection, similar to that observed in mice infected with R20291. Mice infected with FM2.5_varB, in contrast, showed a limited pattern of weight loss, like those animals infected with FM2.5, which stabilized from 48 hpi.

To determine whether these differences were associated with changes in toxin production, the variants were cultured *in vitro* for up to 72 h. Filtered spent growth medium removed from the cultures at 36 and 72 hpi were used to determine the production of toxin B. Both FM2.5_varA and FM2.5_varB produced levels of toxin comparable to R20291 at these time points (Fig 5b), with toxin-mediated damage to Vero cells resulting in cell rounding, cellular loss and reduced

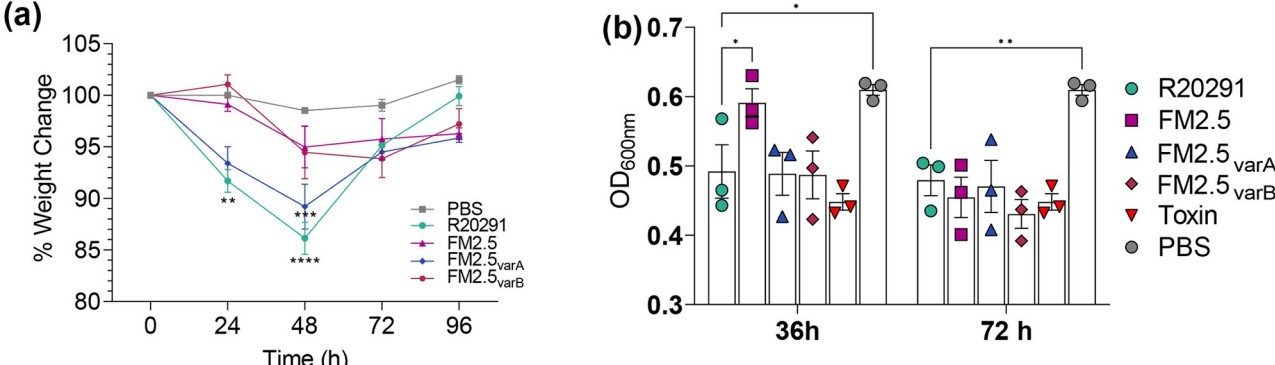

**Fig 5. Functional analysis of FM2.5_varA and FM2.5_varB *in vivo* and *in vitro*.** (**a**) Female C57/Bl6 mice were challenged with spores of R20291, FM2.5, FM2.5_varA, FM2.5_varB or mock infected with sterile PBS. Weight loss was monitored at the same timepoint each day for four consecutive days following infection. Each point shows the average weight change for a minimum of 5 animals per treatment. (**b**) *In vitro* toxin activity as measured through challenge of Vero cells. Samples were prepared by filtering supernatant following growth of *C. difficile in vitro* for 36 or 72 h; activity was measured by the challenge of Vero cells. Supernatants were harvested at the same phase of growth for each strain. Cells treated with PBS or with purified Toxin B were included as negative and positive controls respectively. OD_600 represents the optical density of Giemsa stain incorporated and released from intact Vero cells, hence high OD represents limited toxicity. Results displayed are the mean ± SEM of at least three independent replicates. Statistical tests were conducted using GraphPad Prism software v.12. Statistical significance is indicated: ns—not significant; *p < 0.05; **p < 0.01; and ***p < 0.001.

levels of staining with Giemsa. In agreement with previous reports [20], FM2.5 produced significantly less toxin than R20291 at 36 h, although toxin production levels were comparable in all strains by 72 h.

## Modification of the S-layer results in large changes in gene expression

To gain a greater understanding of the differences in gene expression between R20291, FM2.5 and variant strains, comparative RNA-Seq analysis was conducted following *in vitro* growth. SlpA was the most highly expressed transcript in all strains (R20291, FM2.5, FM2.5$_{varA}$ and FM2.5$_{varB}$) which is unsurprising given that the S-layer is comprised of approximately 600,000 copies of SlpA. In FM2.5, *slpA* is transcribed normally but is not translated to a full-length protein due to the frame-shift early in the open reading frame. Consequently, as transcripts are generated to an equivalent level, this gene does not appear in our list of differentially expressed genes. In contrast, comparative analysis of transcripts generated by R20291 and FM2.5 revealed over 287 differentially expressed genes (DEGs) (Fig 6a), linked to alterations in metabolism, transport, membrane integrity and sporulation (Fig 6b). In contrast, less differences were observed when R20291 was compared to FM2.5$_{varA}$ (44 DEGs), than FM2.5$_{varB}$ (185 DEGs), which showed similar numbers of DEGs to FM2.5. This correlates well with the observed phenotype of these strains within animals—FM2.5$_{varA}$ associated with wild type like disease and FM2.5$_{varB}$ with FM2.5-like attenuation. Analysis of these data suggest that differences in observed disease severity could be linked to changes in transcription of several virulence-associated traits, including toxin A and B, and genes associated with sporulation (Fig 6c). While in the case of FM2.5$_{varA}$, the recovery of an intact S-layer would appear to be sufficient to restore wild type gene transcription, only partial transcription profile restoration, including toxin expression, was observed in FM2.5$_{varB}$. However, as toxin activity was observed at 36 h in culture, the alterations in transcription control seen in FM2.5 and FM2.5$_{varB}$ would appear to be limited to timing rather than absolute prevention of toxin production.

Taken together, these data suggest that expression of the S-layer plays a key role in *C. difficile* gene expression which contributes to disease severity within the host.

## Discussion

The S-layer of *C. difficile* has long been considered as integral to its physiology and pathogenesis, with several roles reported, including adherence to the epithelial barrier [26], immune cell signalling [32, 38], resistance to antimicrobial peptides [20] and sporulation efficiency [20]. Here, we describe the pathogenesis of the S-layer-null mutant FM2.5 within the mouse model of disease and report the recovery of toxin-independent, spontaneous S-layer variants in which SlpA expression is restored. This unexpected but reproducible phenomenon supports the growing evidence that this envelope structure plays a key role in adaptation and survival within the host.

FM2.5, a strain originally selected through its resistance to the engineered R-type bacteriocin Av-CD291.2, was previously reported to be attenuated in the Syrian hamster model of *C. difficile* [20]. These studies indicated that SlpA was essential for disease, with no diarrhoeal symptoms observed in infected animals, despite the recovery of FM2.5 from the caecum and colon of infected hamsters 14 days pi. While we confirmed that the attenuated phenotype was reproducible in mice, we also report the recovery of SlpA variant clones from infected animals. Interestingly, SlpA variants were not observed during hamster infections [20], despite using the same chromogenic agar for recovery of FM2.5 isolates from infected animals. Although we cannot say for certain that such variants did not exist within this population, the failure to observe such clear phenotypic changes suggests that differences in the local environmental

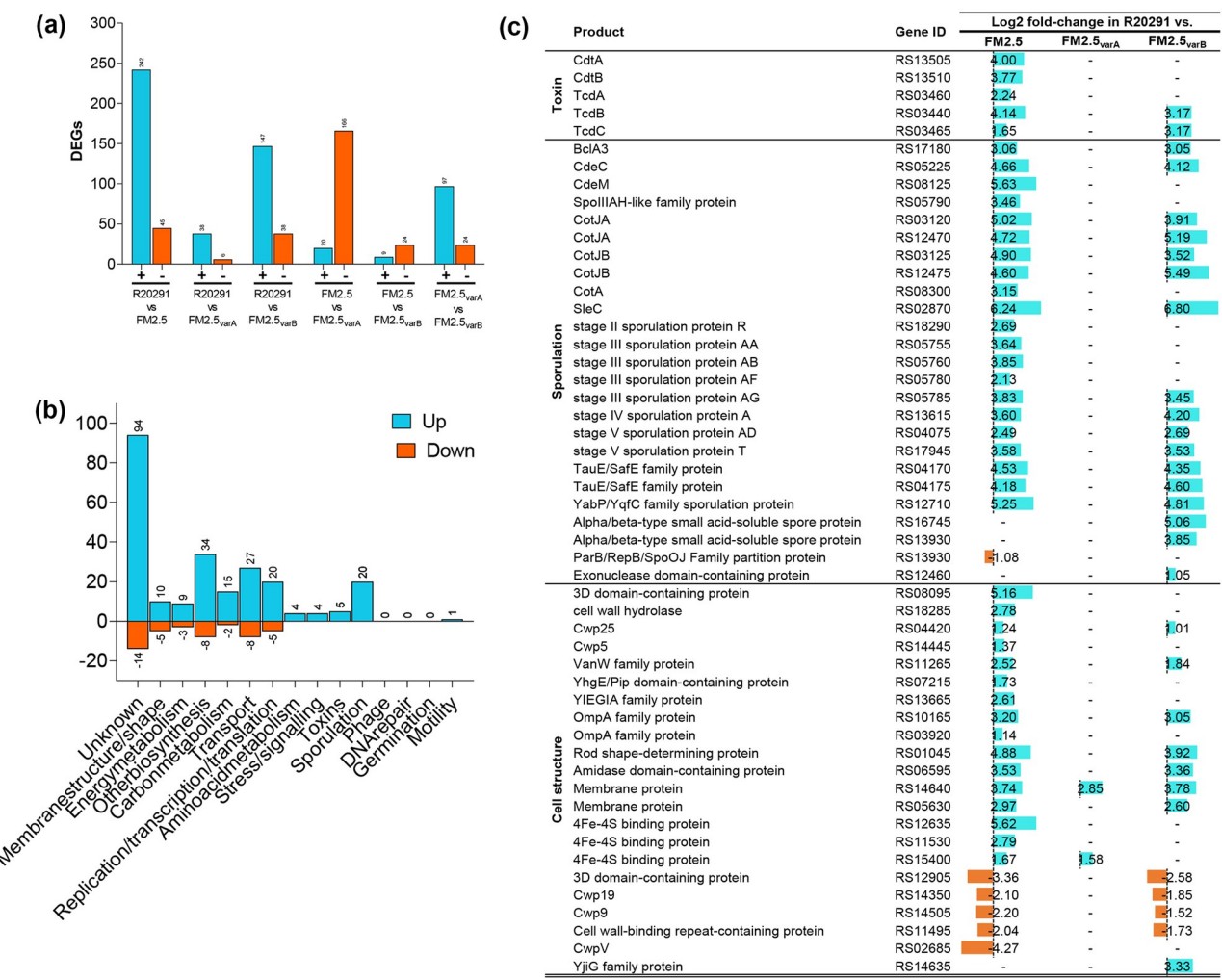

**Fig 6. Global transcriptional differences between isolates of *C. difficile* following *in vitro* growth.** Analysis of mRNA recovered from *in vitro* grown cultures of *C. difficile* isolates R20291, FM2.5, FM2.5$_{varA}$ and FM2.5$_{varB}$. (**a**) Total number of differentially expressed genes (DEGs) between experimental groups are highlighted in turquoise (upregulated) and orange (downregulated). (**b**) DEGs from experimental comparison of R20291 and FM2.5 were categorised based on function. (**c**) Transcriptional differences in select genes of FM2.5, FM2.5$_{varA}$ and FM2.5$_{varB}$ relative to R20291. Lists of all identified differentially expressed genes are provided (S1 Spreadsheet).

conditions within the hamster and mouse may influence the amplification and outgrowth of these S-layer variant strains. The observation that FM2.5 is acutely sensitive to LL-37 and lysozyme [20] and becomes resistant following S-layer restoration [24] further supports the premise that generation of an intact S-layer provides a competitive advantage *in vivo*.

Although SlpA variants of FM2.5 have not been observed *in vitro* during sequential growth and recovery, several factors may influence their presence, albeit at low numbers, within the inoculum used to infect mice. As an obligate anaerobe, the sensitivity of *C. difficile* vegetative cells to oxygen complicates quantification of dosing of animals during oral challenge. In contrast, the preparation and use of spores correlates with the natural route of infection, and additionally avoids co-administration of toxins expressed during *in vitro* growth and preparation of vegetative cells. However, as FM2.5 has a known reduced sporulation efficiency [20], it is possible that any variants expressing an intact S-layer may sporulate more efficiently and therefore represent a higher proportion of the inoculum used for infection. Sequence analysis

of *slpA* of several different batches of FM2.5 spores supported this hypothesis, with a small (< 5%) proportion of the population displaying variations upstream of the original *slpA* mutation. Subsequent selection and amplification of these variants *in vivo* highlights the competitive advantage offered by the S-layer in *C. difficile* intestinal survival. This also provides an explanation for the significantly lower recovery of FM2.5 in mice, at 24 hpi compared to R20291, which becomes equivalent at later time points; reflecting the time required for the variant population to multiply to equivalent numbers within the gut. As germination rates of R20291 and FM2.5 have been previously shown to be equivalent [20], it is unlikely that the difference observed at 24 hpi reflects lower or less efficient rates of germination by FM2.5. Alternatively, lower rates of FM2.5 at 24 hpi could reflect the dynamics between killing of susceptible SlpA-null clones by anti-microbial peptide activity and low numbers of variants within the gut at this time point.

FM2.5 has previously been reported to delay toxin production [20] which, coupled with the apparent delayed growth *in vivo*, could account for the difference in weight loss between R20291 and FM2.5 infected mice after 24 hpi. Using our FM2.5$\Delta PaLoc$ strain, we were able to demonstrate that the weight loss and toxin production are linked. However, this raises the question as to why animals infected with FM2.5, that show high and equivalent levels of tissue colonisation by 48 hpi and which are producing high amounts of toxin, do not show equivalent levels of weight loss and tissue inflammation at this and subsequent time points. Several studies have shown that the S-layer is essential in immune activation [39, 40] driving the production of proinflammatory cytokines via TLR4/MyD88 dependent pathways and enhancing the toxin-activated inflammasome [32, 41, 42]. Together, this implies that the timing or spatial localisation of toxins and the S-layer relative to the epithelial barrier could be crucial to immune activation. This hypothesis is further supported by the observation that FM2.5$_{varB}$, which showed an equivalent reduction in early (6 h of growth) toxin expression to FM2.5 in the RNA-Seq analysis, also displayed reduced disease severity in the mouse.

Regulation of toxin production is highly complex with multiple mechanisms of regulation reported to influence expression in response either directly (CodY, CcpA) or indirectly via interactions with *tcd*R (CodY, RstA, CcpA) in response to various stimuli including nutrient availability and quorum sensing [43]. More recently, c-di-GMP signalling and phase variation via seven invertible switches have been implicated in modulation of expression of genes associated with motility, toxin production and colony morphology [44, 45]. Currently, it is unclear what mechanisms are involved in the observed altered toxin transcripts expression in FM2.5 and FM2.5$_{varB}$ and reduced toxin production by FM2.5 during early growth in culture. However, this does not appear to be linked to phase variation, as orientation of the *flgB* switch, which has been shown to influence toxin expression, does not correlate with virulence observed *in vivo* (S6 Fig). Toxin production was delayed rather than absent in FM2.5 and FM2.5$_{varB}$, with transcription of *tcdB* at a lower level in these strains than in R20291 or FM2.5$_{varA}$. However, equivalent functional activity to R20291 was observed when FM2.5$_{varB}$ was grown for 36 h and FM2.5 for 72 h *in vitro*. This delay rather than impairment of toxin production supports the hypothesis that disease severity might be linked to the timing and co-ordination of S-layer mediated immune signalling and toxin expression within the host.

Mice infected with either FM2.5$_{varA}$ or FM2.5$_{varB}$ showed different disease severity, as indicated by differences in weight loss. The low virulence of an FM2.5$_{varB}$ infection may be, in part, explained by the structural differences. Indeed, structural analysis of SlpA$_{varB}$ revealed a different packing of SlpA molecules in the array, with a rearrangement of the position of SLP$_H$ and both interacting domains (Fig 3). This suggests a considerable degree of adaptability of both SLP$_L$, where the changes are located, and SLP$_H$, to accommodate varying interactions between neighbouring molecules. The absence of density to model D2 in SlpA$_{varB}$ further

illustrates that this domain is dispensable for S-layer assembly, as previously reported [24]. As S-layer assembly is maintained mostly by hydrogen bonds and salt bridges [24], rearrangement of the subdomains to create structures with complementary surface charges seems to enable the different assemblies observed so far. These changes in quaternary structure, with minor changes of secondary and tertiary structure of the subdomains, suggests that the ability to form a paracrystalline array is central to S-layer function and can be achieved in different arrangements. The S-layer must retain a certain degree of flexibility, not only to account for the cell pole curvature and allow cell division, but also for incorporation of minor cell wall proteins that enhance functionality. The presence of a more intricate network of interactions and more extensive interface areas between neighbouring molecules seen in the SlpA$_{varB}$ structure, when compared to the R20291-related SlpA$_{R\Delta D2}$ structure (S2 Table), suggests a potentially less flexible paracrystalline array. This could also reduce the ability to incorporate specific functions of minor CWPs, which may help explain differences in disease patterns observed between SlpA$_{varB}$ and the other variant strain, SlpA$_{varA}$. Further structural studies of SlpA$_{varA}$ and SlpA$_{R20291}$ as well as detailed analysis of S-layer assembly and composition, including the capacity to incorporate other minor cell wall proteins, will help elucidate the role of specific aspects of the S-layer.

While identification of SlpA variants was unexpected and adds complexity to the interpretation of the data from the mouse disease model, the rapid recovery of these strains highlights the key contribution that the expression of an intact S-layer offers to *C. difficile* infection *in vivo*. This work supports previous observations that strains lacking the S-layer are less virulent *in vivo*, although it remains difficult to identify the specific contribution of the S-layer in the infection process. Instead, this work highlights the potential multifunctional contribution that the S-layer plays in disease as, despite the number of differentially expressed genes observed between R20291, FM2.5 and the variants *in vitro*, recovery of the intact S-layer was sufficient to restore virulence, at least in one variant. Importantly, isolation and characterisation of these variants, together with greater knowledge of gene regulation and metabolic pathways impacted, offers a new opportunity to better understand the structural and functional role of the S-layer in *C. difficile* pathogenesis.

## Materials and methods

### Bacterial strains and growth conditions

The bacterial strains used in this study include *C. difficile* strain R20291, its derivative FM2.5 [20], FM2.5$_{varA}$, FM2.5$_{varB}$, FM2.5$_{varC}$, FM2.5Δ*PaLoc* and FM2.5Δ*PaLoc*$_{varD}$ (this study). Strains were routinely grown under anaerobic conditions on Braziers cycloserine, cefoxitin egg yolk (CCEY) agar (Oxoid, UK); CHROMID *C. difficile* Chromogenic medium (BioMérieux); or in Tryptone yeast (TY) broth (Oxoid, UK).

### Generation of FM2.5Δ*PaLoc*

Homologous recombination was used to generate a derivative of strain FM2.5 that lacked the entire pathogenicity locus (PaLoc). Briefly, 1.2 kb up and downstream of the PaLoc was amplified by PCR using oligonucleotides RF920 and RF921, and RF922 and RF923 (S3 Table), respectively, and cloned by Gibson assembly into plasmid pJAK112 [46] that had been linearised by PCR using RF311 and RF312. The resulting plasmid, pJAK143, was then conjugated into *C. difficile* [47] and mutagenesis to knock out the PaLoc was carried out using standard allele exchange [4].

## Murine model of infection

All procedures were performed in strict accordance with the Animals (Scientific Procedures) Act 1986 with specific approval granted by the Home Office, UK (PPL 60/8797 and PPL PI440270). These applications were also considered and approved by the University of Glasgow Animal Welfare Ethical Review Body (AWERB). Food and water were provided *ad libitum* and animals kept at a constant room temperature of 20–22 ˚C with a 12 h light/dark cycle. Groups of up to six C57/bl6 female mice aged 6–8 weeks supplied by Charles River (Edinburgh) were used in each treatment group. All animals were screened for colonisation with *C. difficile* in advance of the start of each experiment by plating faeces on ChromID selective media (Biomeuriex). An antibiotic cocktail (kanamycin, 0.40 mg ml$^{-1}$; metronidazole, 0.215 mg ml$^{-1}$; colistin, 850 U ml$^{-1}$; gentamicin, 0.035 mg ml$^{-1}$ and vancomycin, 0.045 mg ml$^{-1}$ [all Sigma Aldrich, UK]) was administered *ad libitum* in the drinking water as previously described [35] with clindamycin sulphate (150 mg Kg$^{-1}$), administered by oral gavage following cessation of the antibiotic cocktail. Animals were each challenged with approximately $10^6$ spores of *C. difficile* 72 h after clindamycin treatment. Mice were monitored closely post-infection and weighed daily to determine the severity of the disease. Animals with a weight loss greater than 10% of pre-challenge weight were given soft food and were culled if weight loss reached 20%.

## *C. difficile* shedding and organ colonization

Fresh faecal samples collected daily were weighed, serially diluted in phosphate buffered saline (PBS) and cultured on CCEY agar at 37 ˚C for 48 h. At the experimental endpoint, animals were culled, and the caecum and colon harvested. Enumeration of total counts and spore-specific counts in lumen associated material were performed as previously described [35]. In brief, total viable counts were determined by plating serial dilutions on ChromID selective media. Spores were enumerated following standard heat treatment of the samples at 65 ˚C for 20 min [33].

## Quantification of toxin expression

Quantification of toxin activity was performed using monolayers of Vero cells (kidney epithelial cells) as described previously [48]. Briefly, toxin was recovered from the spent filtered TY medium used to support bacterial growth for 36–72 h. Spent medium was recovered at the same stage of the growth cycle and at the same $OD_{600nm}$. *In vivo* toxin activity was measured by filtering using a 0.2 μm filter, luminal content collected from the caecum and colon of infected mice. Luminal contents collected from the caecum and colon of uninfected mice served as negative controls. Samples for toxin measurement were tested by the addition of serial dilutions to confluent monolayers within 72 h of collection. Cells and toxin were co-cultured for 24 h before cells were washed with phosphate buffered saline (PBS), fixed with 5% formal saline (Fisher), and stained with Giemsa for 30 min, before thorough washing to remove excessive stain. For data presented in Fig 1d and S1b and S5c Figs, toxin activity was determined as the reciprocal of last dilution in which toxin activity was observed, i.e., showing cell destruction. In Fig 5, to quantify the toxin activity more precisely, excess stain was removed by washing with PBS before cells were permeabilised to release internalised stain using 200 μl 1% SDS. 100 μl of the supernatant was transferred to a fresh plate and the $OD_{620nm}$ values determined. In this context, higher values indicate Vero cells are intact and unaffected by the toxin, lower values indicate that toxin mediated damage prevents uptake and retention of the dye. Cells treated with PBS only or with 0.0005 μg ml$^{-1}$ purified Toxin B (Sigma, UK) were included as negative and positive controls respectively.

## Histology and immunohistochemistry

Tissue samples were harvested from the caecum and colon of antibiotically susceptible animals infected with either R20291 or FM2.5 at post-mortem, 48 hpi. These tissues were gently washed in sterile PBS and immediately fixed in 10% formalin. Embedded tissue sections were cut and stained with Hematoxylin and Eosin [35]. Blind histological scoring of tissue was performed on three independent sections of caecal and colonic tissue from each mouse. Each section was scored out of a total of 15, with a score of 0 indicating no change, 1 slight change, 2 moderate change and 3 significant change, for the following categories: epithelial damage, neutrophil migration, haemorrhagic congestion, tissue oedema and crypt hyperplasia. Data presented represents the mean scores calculated for four mice for caecal, and three mice for colon assessment for each treatment.

## *slpA* sequencing from isolated variant clones

Individual clones of bacteria, recovered from faecal or tissue associated material that showed different morphology on ChromID plates, where subject to at least two rounds of clonal selection. Genomic DNA was isolated from a 20 ml culture grown anaerobically for 18 h in tryptic soya broth (TSB). Bacterial cells were initially disrupted enzymatically by resuspending the pellet in lysis buffer (20 mM Tris-Cl, pH8.0, 2 mM Na EDTA, 1.2% Triton X-100, lysozyme 200 mg ml$^{-1}$), and incubated at 56 ˚C for 90 min. The DNA was recovered using the DNeasy Blood and Tissue Kit (Qiagen), following manufacturer's instructions. A 478 bp fragment of *slpA*, centred on the FM2.5 point mutation [20], was amplified by using oligonucleotides RF110 and RF111 (S3 Table). The resulting fragments were subjected to Sanger sequencing and compared to wild type and FM2.5 sequences.

## Isolation and sequencing of *slpA* from faecal extracts

Faecal samples were also collected for unbiased sequencing of *slp*A within the population, by directly extracting DNA from faecal samples using the FastDNA SPIN kit for soil (MP Biomedicals). Briefly, approximately 200–600 mg of faeces sample were suspended in 978 μl sodium phosphate buffer with 122 μl MT buffer lysis solution. Samples were then homogenised in a FastPrep instrument using two 30 second pulses, at speed setting 6.5, in lysing matrix E. Samples were centrifuged for 10 min at 14,000 *x g* to remove debris. 250 μl protein precipitation solution was added to the lysate supernatant, and the precipitant formed removed by centrifugation at 14,000 *x g* for 5 min. DNA was then bound to a silica matrix, washed using the kit wash buffer, and eluted with water. For analysis of *slpA* sequences in spores used for challenge experiments in mice, samples were treated with 0.1% taurocholic acid for 60 min to induce germination and then boiled for 5 min in the presence of Chelex resin to release DNA. DNA extracted from either faeces or germinated spores was used as a template for PCR amplification of a 330 bp fragment of *slpA* using Phusion polymerase (NEB) and oligonucleotides RF2193 and RF2194 (S3 Table). Resulting DNA fragments were purified and sequenced using the amplicon-EZ service offered by GENEWIZ (Azenta Life Sciences).

## Extraction and western immunoblot analysis of S-layer and associated proteins

Surface layer proteins were extracted using low pH glycine as previously described [49] and analysed by SDS-PAGE using standard methods [50] (Fig 2 and S3 Fig). Proteins were transferred to nitrocellulose membranes via semi-dry transfer (Bio-Rad Trans Blot Turbo; 25 V, 30

min) for western immunoblot analysis. Transfer efficiency was confirmed by PonceauS staining of membrane post transfer, and Coomassie staining of the polyacrylamide gel following transfer. Membranes were blocked for 1 h in Phosphate-buffered Saline containing 0.1% Tween20 (PBS-T) with 5% milk powder. Blots were subsequently incubated in primary antibody (rabbit anti-SLP$_H$ raised against *C. difficile* 630 1:100,000 dilution; rabbit anti-SLP$_L$ raised against *C. difficile* R20291 1:200,000 dilution) in PBS-T containing 3% milk powder, for 1 h at room temperature. Membranes were washed thoroughly in PBS-T before incubation with secondary antibodies (anti-rabbit horseradish peroxidase, Promega WB401B 1:2,500 dilution) for 1 h at room temperature. Blots were washed in PBS-T before detection by chemiluminescence (Bio-Rad). Molecular weight (MW) markers (Thermo Scientific 26616) were imaged (Bio-Rad ChemiDoc XRS+) simultaneously and overlaid onto the blots to aid visualisation. Original full blots are shown in S3 Fig.

## Whole genome sequence analysis

Genomic DNA recovered from R20291, FM2.5, FM2.5$_{varA}$, FM2.5$_{varB}$ were sequenced by MicrobesNG (Birmingham UK) using Illumina based technologies. The raw data was then aligned to the reference genome (NC_013316.1) using Bowtie2. Polymorphisms were called using FreeBayes under default settings and VCF file was filtered to include only mutations identified with a read depth of > 50 and in at least 50% of reads. To build the tree (S2 Fig), the filtered mutations across all strains were merged and converted into a bed file. Regions were expanded by 100 bp on each side and these were considered the mutable regions. The phylogenetic tree was constructed using PhyloTree (https://github.com/samandmac/PhyloTree), which uses BLAST and PHYML to generate a tree in Newick format, and subsequently an R script which allows visualisation of the tree using the ggtree package. The raw sequence data has been deposited to the Gene Expression Omnibus; reference ID GSE205747. Note: file name Rv9 contains the genomic data for FM2.5$_{varA}$, and RV189 for FM2.5$_{varB}$ respectively.

## Protein purification and X-ray crystallography

*C. difficile* variant strains were cultured in 400 ml of TYG broth for 16 h. Cultures were then centrifuged at room temperature at 4,696 x *g* and resulting pellets were washed with 40 ml of 0.01 M HEPES pH 7.4 and 0.15 M sodium chloride (HBS) buffer. S-layer extraction was performed by resuspending the washed pellet in 4 ml of 0.2 M glycine-HCl pH 2.2 and centrifugation for 5 min at 21,100 x *g*. Collected supernatant was then neutralized with 2 M Tris-base. S-layer extract was filtered and resolved onto a Superdex 200 26/600 column using an ÄKTA Pure FPLC system (Cytiva) in 50 mM Tris-HCl pH 7.5, 150 mM NaCl buffer.

Purified SlpA$_{varB}$ at 10 mg ml$^{-1}$ was subjected to crystallization using a Mosquito liquid handling robot (TTP Labtech), with the sitting drop vapor-diffusion method, at 20 ˚C. Crystals were obtained in 0.03 M magnesium chloride hexahydrate; 0.03 M calcium chloride dihydrate, 0.12 M ethyleneglycol, 0.05 M Tris (base); 0.05 M bicine pH 8.5, 20% v/v glycerol; 10% w/v PEG 4,000.

Data were collected on the I24 (λ = 0.71 Å) beamline at the Diamond Light Source Synchrotron (Didcot, UK; mx24948-136) at 100 K. Two datasets were collected from one crystal and initially processed by the automatic multi-crystal data-analysis software pipeline xia2.multiplex [51] within the Information System for Protein Crystallography Beamline (ISPyB), re-processed using Automatic Image Processing with Xia-2 (DIALS [52] and Aimless 3d [53]) and scaled with Aimless within ccp4.cloud of CCP4 [54] software suit.

The initial model of the core SLP$_H$ was obtained by molecular replacement in Phaser [55], using an SlpA$_{varB}$ model of CWB2 domains, derived from the SlpA$_{RΔD2}$ model (PDB ID:

7ACZ) and calculated using SWISS-MODEL [56]. The generated solution model was then subjected to automatic model building with Modelcraft [57], followed by manual building with Coot [58] and refinement in Refmac5 [59].

Final models were obtained after iterative cycles of manual model building with Coot and refinement in phenix_refine [60]. Data collection and refinement statistics are summarized in S1 Table.

PDBePISA [61] was used to investigate interdomain and protein-protein interfaces in the crystallographic lattice to identify interacting residues, which were confirmed by manual inspection within COOT.

Structural representations were generated using PyMOL Molecular Graphics System (Schrödinger, LLC).

## Protein structure prediction

Homology models for $SlpA_{R20291}$ and $SlpA_{varA}$ were generated by providing SWISS-MODEL webserver with the $SlpA_{varB}$ (PDB ID: 8BBY) or $SlpA_{R\Delta D2}$ (PDB ID: 7ACZ) as user templates, as well as without a template. Structural alignments between predicted models and templates were performed using COOT [58]. As $SlpA_{R\Delta D2}$ lacks the D2 domain, predicted models based on this experimental model have a disordered D2 domain. Therefore, overall comparison of the three predicted models was based on models calculated in the default mode, which uses $SlpA_{R7404}$ (PDB ID: 7ACX) as a template, while analysis of the modification-containing region in the D1 domain was done using $SlpA_{R\Delta D2}$ or $SlpA_{varB}$ derived models.

## Recovery of mRNA for RNA sequence analysis

RNA was recovered from *C. difficile* strains R20291, FM2.5, $FM2.5_{varA}$ and $FM2.5_{varB}$ which had been cultured *in vitro* in TY broth. Briefly, bacterial cells reaching an $OD_{620nm} = 0.6$ were pelleted (5000 x *g*, 15 min) and immediately fixed in 1.5 ml RNA-protect (Qiagen) for 10 min before being processed using a PureLink RNA mini kit (Ambion) to extract total RNA. To ensure maximal lysis of bacteria and recovery of RNA, the bacterial pellet was additionally subject to treatment with 100 ml lysozyme solution (10 mg ml$^{-1}$ in 10 mM Tris-HCl, pH8.0, 0.1 mM EDTA), 0.5 ml 10% SDS solution and 350 ml of Lysis Buffer (Invitrogen PureLink RNA Mini Kit) containing 2-mercaptoethanol. Cells were then homogenised using MP Biomedical beads (0.1 mm) and bead beater (MP Biomedical FastPrep24) with cells subject to 2 cycles of 60 s beating, followed by incubation on ice for 2 min. Total RNA was then extracted using the standard PureLink RNA mini kit protocol, according to the manufacturer's specifications. Genomic DNA was removed using a TURBO DNase kit (Ambion) and samples were tested for efficient removal of DNA by conventional PCR. Samples for RNA-Seq were prepared in triplicate on two separate occasions (6 samples for each) for all four bacterial strains.

Illumina library preparation of mRNA samples for RNA-Seq was prepared using a TruSeq Stranded mRNA library prep kit (Illumina) according to the manufacturer's instructions. Sequencing was performed on the Illumina NextSeq 500 platform (75 bp length; single-end). Library generation, optimisation of amplification and sequencing were performed at the University of Glasgow Polyomics facility. Quality control of sequencing data was performed using FastQC (Babraham Bioinformatics) to assess the minimum Phred threshold of 20 and potential data contamination. The raw data has been deposited to the Gene Expression Omnibus reference ID GSE205747. Note: files with sample numbers 1–6 reflect samples generated and analysed independently for each strain; Rv9 was the original name of $FM2.5_{varA}$, and RV189 for $FM2.5_{varB}$, respectively.

## RNA analysis and identification of differentially expressed genes

Raw RNA-Seq datasets were subject to the following pipeline. Firstly, fastQ files were assessed using FastP [62] and then were aligned to the *C. difficile* R20291 (accession number NC_013316) reference genome using STAR [63] (v2.6) with–quantMode GeneCounts–outFilterMultimapNmax 1 and–outFilterMatchNmin 35. We used a Star index with a–sjdbOverhang of the maximum read length − 1. Next, read count files were merged and genes with mean of < 1 read per sample were excluded. Finally, the expression and differential expression values were generated using DESeq2 [64] (v1.24). For differential comparisons, we used an A versus B model with no additional covariates. All other parameters were left to default.

The processed data was then visualised using Searchlight [65], specifying one differential expression workflow for each comparison, an absolute $log_2$-fold cut-off of 1 and adjusted *p* of 0.05. All other parameters were left to default.

## Pathway analysis methods

Functional and metabolic pathways were implied by interrogation of the WP numbers assigned using the *C. difficile* annotated genome (NC_013316.1) and the Uniprot or NCBI Blastp databases. Gene ontology (GO) was assigned based on Biological Process assignment within Uniprot. Comparative analysis of all differentially expressed genes are provided (S1 Spreadsheet).

## Analysis of phase variable switch orientation

To analyse the orientation of the seven *C. difficile* R20291 phase variable switches, orientation specific qPCR was carried out as previously described [44]. All strains were grown on BHI agar and mixed populations were prepared by pooling 100–200 colonies of each followed by genomic DNA purification using standard phenol-chloroform extraction and isopropanol precipitation. qPCR was performed using SYBR Green JumpStart Taq following the manufacturer's instructions with 100 ng gDNA and 100 nM of each oligonucleotide (S3 Table). Reactions were run in a Bio-Rad CFX Connect RT-PCR instrument using the following parameters: 95 ˚C for 3 min, followed by 40 cycles of 95 ˚C for 10 sec, 60 ˚C for 15 sec, 72 ˚C for 15 sec and 64 ˚C for 10 sec, followed by a melting curve: 95 ˚C for 10 sec, 55 ˚C for 5 sec and 95 ˚C for 5 sec. Products were proportionally quantified using the *rpoA* gene as a reference and data presented as the percentage of the switch in the orientation of the published R20291 reference genome [66].

## Statistical analysis

Statistical analysis was carried out in GraphPad Prism v.12. Tests used included for weight loss a two-way ANOVA with Tukey post test; for bacterial counts and toxin measurements, a two way ANOVA; for tissue histology analysis an ordinary one-way ANOVA. Statistical significance is indicated: ns—not significant; *p < 0.05; **p < 0.01; and ***p < 0.001.

## Supporting information

**S1 Fig. Characterisation of *slpA* deficient *C. difficile* in the colon of infected mice.** Female C57/Bl6 mice were challenged with spores of R20291 (green) or FM2.5 (purple), or mock infected with sterile PBS (grey). (**a**) CFU ml$^{-1}$ of total (clear fill) bacterial recovery or spores (fill pattern) in colonic contents at 24, 48 and 96 hpi (n = 5 at each time point except R20291 at 24 hpi; n = 4). (**b**) Toxin activity within colonic content at 24, 48 and 96 hpi; through challenge of Vero cells *in vitro* (n = 5 at each time point). Results displayed indicate the reciprocal of lowest dilution at which toxin activity could be measured. (**c**) Histological scoring of sections of

the upper (clear fill) and lower (fill pattern) colon sections from mice challenged with R20291, FM2.5 and PBS treated animals. Results displayed are the mean ± SEM of assessment of at least three regions of tissue from at least three individual animals. (**d**) Histopathological sections representing caecal (i, ii and iii) sections following challenge with PBS (i); R20291 (ii); or FM2.5 (iii). Scale bars represent 100 μm. Statistical tests were conducted using GraphPad Prism software v.12. Statistical significance is indicated: ns—not significant; *p < 0.05; **p < 0.01; and ***p < 0.001.
(TIF)

**S2 Fig. Confirmation of the close relationship between R20291, FM2.5 and *slpA* variants, FM2.5$_{varA}$ and FM2.5$_{varB}$ using whole genome sequence analysis.** Chromosomal DNA was recovered from each strain and the raw data was aligned to the reference genome (NC_013316.1). (**a**) The variants were identified as closely related to both FM2.5 and R20291 using whole genomic sequencing to generate phylogenetic trees based on identification of polymorphisms identified during comparative analysis of sequences. Mutations were only included with a read depth of > 50 and which was present in at least 50% of reads. (**b**) Evaluation of the *slpA* sequence from all four strains from genomic data confirmed mutations were limited to the regions highlighted above. Mutations originally identified by PCR analysis were confirmed by short read genomic sequencing of individual strains. Insertions of the previously identified additional A in FM2.5, FM2.5$_{varA}$, FM2.5$_{varB}$ are highlighted within the purple box, deletion of T in FM2.5$_{varA}$ by the dark blue box and insertion of CTTAG in FM2.5$_{varB}$ in red. Grey indicates the depth of sequence read coverage.
(TIF)

**S3 Fig. Raw images of SDS-PAGE gels and western immunoblots shown in Fig 2d, 2e and 2f.** (**a**) SDS-PAGE of S-layer extracts, non-boiled and boiled prior to resolving on a polyacrylamide gel. Molecular weight marker (M, PageRuler Prestained Protein Ladder, ThermoScientific); R20291 (1); FM2.5 (2); FM2.5$_{varA}$ (3); FM2.5$_{varB}$ (4). For Fig 2d, the SDS-PAGE gel was cropped to show only heated samples. (**b**) To assess efficiency of electrotransfer, the nitrocellulose membrane was stained with PonceauS (Sigma) and (**c**) polyacrylamide gel post-transfer was stained with Coomassie. The Western immunoblot analysis was performed comparing (**d**) non-boiled and (**e**) boiled S-layer extracts. For Fig 2e, the blots were cropped to allow focus on the heated samples only. Throughout all images, the SLP$_L$ is indicated with blue arrowhead and SLP$_H$ is indicated with yellow arrowhead. The additional band, detected with anti-SLP$_H$ CD630 antibodies and highlighted with magenta arrowhead, corresponds to SLP$_H$ missing interacting domain HID [24]. The high molecular weight band detected with anti-SLP$_L$ R20291, indicated with green arrowhead, could correspond to an apparent SDS-resistant oligomers of SLP$_L$ or non-specific binding of the antibody to CwpV, a phase variable protein.
(TIF)

**S4 Fig. Tiling of SlpA$_{varB}$ and comparison with predicted models of SlpA$_{varA}$ and SlpA$_{R20291}$.** (**a**) Tiling of SLP$_H$ CWB2 motifs is stabilised by interacting domains. Poisson-Boltzmann electrostatic potential calculated for SlpA$_{varB}$ SLP$_H$, represented as a charge distribution (positive—blue; negative—red) on the surface representation of SLP$_H$ array. Interacting surfaces between molecules 1–2, defined by pseudo-symmetry related CWB2$_3$-CWB2$_1$, and between molecules 1–3, defined symmetry-related CWB2$_3$ triangular prism faces, are labelled. The cavity between symmetry-related CWB2$_1$-CWB2$_2$ surfaces, (117 Å, green arrow, left) is partially occluded by SLP$_L$ D1 (crimson) and by the insertion of the LID/HID domains

(electrostatic potential surface representation), as shown on the right panel. A long cavity of ~83 Å at the $CWB2_2$ vertices represented by purple arrow (left) is also occluded by LID/HID domains and interacting $SLP_L$ molecules (bottom). Neighbouring $CWB2_1$-$CWB2_3$ triangular prism faces form the interaction surface, while neighbouring $CWB2_3$ vertices complete the $SLP_H$ tiling. Insertion of the interacting domains bridges neighbouring $SLP_H$ tiles. Interactions across the lattice are maintained via complementary charged interfaces. (**b**) Predicted structural models for $SlpA_{R20291}$ ($SLP_L$—green, $SLP_H$—grey) and $SlpA_{varA}$ ($SLP_L$—dark blue, $SLP_H$—grey) show the same overall structure of both SLPs as the experimental structural model for $SlpA_{varB}$ ($SLP_L$—gold, $SLP_H$—slate blue) and $SlpA_{R\Delta D2}$ ($SLP_L$—gold, $SLP_H$—grey). Left: Cartoon representation of the complex, as seen from the environmental side, with all models superimposed on $SLP_L$ from $SlpA_{varB}$. The different orientation of $SLP_L$ and interaction domains observed in $SlpA_{varB}$ is evidenced by the change in position of $SLP_H$ in this structure when compared to $SlpA_{R\Delta D2}$ and the predicted models, calculated based on $SlpA_{R7404}$ (PDB ID: 7acx). Right: Zoom view comparing D1 region of $SlpA_{varB}$ (crimson) with $SlpA_{R20291}$ (top right, green) and $SlpA_{varA}$ (bottom right, dark blue). Sequence of $\alpha2_L$ in the different variants, with paler colours indicating differences compared to R20291, is shown schematically and corresponding sidechains are represented as sticks in the cartoon representation. Carbon —coloured according to molecule as in (**a**), oxygen—red, nitrogen—dark blue.
(TIFF)

**S5 Fig. *In vivo* challenge of mice infected with FM2.5Δ*PaLoc*.** Female C57/Bl6 mice were challenged with spores of R20291 (green), FM2.5 (purple) or FM2.5Δ*PaLoc* (blue). (**a**) Toxin activity of caecal and colonic content was determined as the reciprocal of last dilution in which cytoplasmic changes to challenged were observed. Statistical significance is indicated: ns—not significant. (**b**) At the experimental endpoint (96 hpi), total *C. difficile* counts were enumerated by colony counts from the caecum and colon contents from individual animals (n = 5 per group). (**c**) Spores were obtained from caecum and colon material and enumerated by colony counts. Vegetative cells were killed by heating samples at 65 ˚C for 20 min prior to plating to ensure only spores remained.
(TIF)

**S6 Fig. Orientations of the phase variable invertible switches in R20291, FM2.5, FM2.5$_{varA}$ and FM2.5$_{varB}$.** To determine if differences in pathogenesis could be linked to a dominance of switch orientations of the 7 known phase switches in *C. difficile*, their orientation in all four strains was evaluated using qPCR, using the method described in [44]. This shows that the differences in virulence observed between FM2.5$_{varA}$ and FM2.5$_{varB}$ does not appear to link to orientation of these switches. Mixed populations representing 100–200 independent colonies were pooled, genomic DNA was extracted and the orientations of the seven switches was determined using a previously described orientation-specific qPCR assay. The gene immediately downstream of the switch and presumed or confirmed to be controlled by the switch, is indicated above each graph. The percentages of each switch found to be in the same orientation as in the published R20291 reference genome are shown. Measurements were performed in duplicate with the mean indicated by the horizontal line. Although variation in orientation between strains is observed for the switches upstream of *pdcC*, *flgB* and *cwpV* these differences do not correlate with observed differences in virulence.
(TIF)

**S1 Table. Data collection and refinement statistics.** Data collection and refinement statistics for structure determination of $SlpA_{varB}$. Data collected at beamline I24, Diamond Light Source. Values in parenthesis correspond to the highest resolution shell. R-work = $\Sigma$ ||Fobs|—|Fcalc|| /

Σ |Fobs| R-free = Σ ||Fobs|—|Fcalc|| / Σ |Fobs| for 5% reflections excluded from refinement I/ sigmaI = I / σ(I) CC1/2 = Σ (Fobs1 * Fobs2*) / (√(Σ |Fobs1|^2) * √(Σ |Fobs2|^2)).
(DOCX)

**S2 Table. Interactions across S-layer tiling.** List of interactions between neighbouring molecules within the S-layer tiling for SlpA$_{varB}$ and previously determined SlpA$_{CD630}$ and SlpA$_{R\Delta D2}$. Interacting atoms and the corresponding residue, together with the bond distance and interaction type (H—hydrogen bond; S—salt bridge) are listed, as determined in PISA and verified in each model using Coot (see material and methods for details). Numbers in parenthesis refer to the number of the neighbouring molecule, as defined in Fig 3b. L—SLP$_L$; H—SLP$_H$. Interactions seen in SlpA$_{varB}$ not previously described in other models are highlighted in crimson.
(DOCX)

**S3 Table. Oligonucleotides used in this study.** Oligonucleotide primers used for PCR and qPCR. Sequences in lowercase indicate homology regions for Gibson assembly. All oligonucleotides were purchased from Eurofins Genomics Europe.
(DOCX)

**S1 Spreadsheet. Pairwise comparative analysis of differentially expressed genes (DEG's) between R20291 and FM2.5, FM2.5$_{varA}$ and FM2.5$_{varB}$.** For differential comparisons, we used an A (R20291) versus B (FM2.5, FM2.5$_{varA}$, FM2.5$_{varB}$) model with no additional covariates. Differential expression values were generated using DESeq2 [64] (v1.24) and the processed data visualised using Searchlight [65]. DEG's were determined as those genes with an absolute log$_2$-fold cut-off of 1 (positive values for relative upregulated genes and negative values for relative downregulated genes) and -adjusted *p* value of 0.05. Functional and metabolic pathways were implied by interrogation of the WP numbers assigned using the *C. difficile* annotated genome (NC_013316.1). Data is also shown for comparisons between FM2.5 vs FM2.5$_{varA}$, FM2.5 vs FM2.5$_{varB}$ and FM2.5$_{varA}$ vs FM2.5$_{varB}$.
(XLSX)

## Acknowledgments

The authors would like to thank Dr Arnaud Baslé, facility manager of the Newcastle Structural Biology Lab, and beamline staff at I24, Diamond Light Source (BAG mx24948) for support on structural data collection and Dr Sam McAllister for help with phylogenetic tree analysis. The contents of this work are solely the responsibilities of the authors and do not reflect the official views of any of the funders, who had no role in study design, data collection, analysis, decision to publish, or preparation of the manuscript.

## Author Contributions

**Conceptualization:** Michael J. Ormsby, Filipa Vaz, Joseph A. Kirk, Anna Barwinska-Sendra, Paola Lanzoni-Mangutchi, Paula S. Salgado, Robert P. Fagan, Gillian R Douce.

**Data curation:** Michael J. Ormsby, Filipa Vaz, Joseph A. Kirk, Anna Barwinska-Sendra, Paola Lanzoni-Mangutchi, John Cole, Roy R. Chaudhuri, Paula S. Salgado.

**Formal analysis:** Filipa Vaz, Joseph A. Kirk, Anna Barwinska-Sendra, Paola Lanzoni-Mangutchi, John Cole, Roy R. Chaudhuri, Paula S. Salgado, Robert P. Fagan, Gillian R Douce.

**Funding acquisition:** Paula S. Salgado, Robert P. Fagan, Gillian R Douce.

**Investigation:** Michael J. Ormsby, Filipa Vaz, Joseph A. Kirk, Anna Barwinska-Sendra, Jennifer C. Hallam, Paola Lanzoni-Mangutchi, John Cole, Roy R. Chaudhuri, Paula S. Salgado, Robert P. Fagan, Gillian R Douce.

**Methodology:** Michael J. Ormsby, Filipa Vaz, Joseph A. Kirk, Anna Barwinska-Sendra, Jennifer C. Hallam, Paola Lanzoni-Mangutchi, John Cole, Roy R. Chaudhuri, Paula S. Salgado, Robert P. Fagan, Gillian R Douce.

**Project administration:** Paula S. Salgado, Robert P. Fagan, Gillian R Douce.

**Resources:** Michael J. Ormsby, Filipa Vaz, Joseph A. Kirk, Anna Barwinska-Sendra, Jennifer C. Hallam, Paola Lanzoni-Mangutchi, Roy R. Chaudhuri, Paula S. Salgado, Robert P. Fagan, Gillian R Douce.

**Software:** John Cole, Roy R. Chaudhuri, Paula S. Salgado, Robert P. Fagan, Gillian R Douce.

**Supervision:** Paula S. Salgado, Robert P. Fagan, Gillian R Douce.

**Validation:** Michael J. Ormsby, Filipa Vaz, Joseph A. Kirk, Anna Barwinska-Sendra, Jennifer C. Hallam, Paola Lanzoni-Mangutchi, John Cole, Roy R. Chaudhuri, Paula S. Salgado, Robert P. Fagan, Gillian R Douce.

**Visualization:** Michael J. Ormsby, Filipa Vaz, Joseph A. Kirk, Anna Barwinska-Sendra, Paola Lanzoni-Mangutchi, John Cole, Paula S. Salgado, Robert P. Fagan, Gillian R Douce.

**Writing – original draft:** Michael J. Ormsby, Filipa Vaz, Joseph A. Kirk, Anna Barwinska-Sendra, Paola Lanzoni-Mangutchi, John Cole, Roy R. Chaudhuri, Paula S. Salgado, Robert P. Fagan, Gillian R Douce.

**Writing – review & editing:** Michael J. Ormsby, Filipa Vaz, Joseph A. Kirk, Anna Barwinska-Sendra, Paola Lanzoni-Mangutchi, John Cole, Paula S. Salgado, Robert P. Fagan, Gillian R Douce.

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
