## [Decision Letter · Decision Letter 0]

2 Jan 2023

Dear Dr. Douce,

Thank you very much for submitting your manuscript "An intact S-layer is advantageous to Clostridioides difficile within the host." for consideration at PLOS Pathogens. As with all papers reviewed by the journal, your manuscript was reviewed by members of the editorial board and by independent reviewers. In light of the reviews (below this email), we would like to invite the resubmission of a significantly-revised version that takes into account the reviewers' comments.

This manuscript was reviewed by three experts in Clostridioides difficile biology. Enthusiasm for the manuscript varied among these Reviewers, but even the most positive Reviewer identified some important concerns (as described in their Reviews). Some of the identified issues might be easily fixed, e.g., terminology issues for the “revertant” strains that are the focus of the study. Other important issues appear likely to require additional experimentation, e.g., concerns regarding statistical rigor/analyses/presentation and experimental design. However, addressing some critical concerns would clearly require additional experimental work, e.g., evaluating whether the suppressor mutants have other genetic or epigenetic differences with phenotypic consequences. 

We cannot make any decision about publication until we have seen the revised manuscript and your response to the reviewers' comments. Your revised manuscript is also likely to be sent to reviewers for further evaluation.

Sincerely,

Bruce A. McClane

Academic Editor

PLOS Pathogens

Michael Wessels

Section Editor

PLOS Pathogens

Kasturi Haldar

Editor-in-Chief

PLOS Pathogens

orcid.org/0000-0001-5065-158X

Michael Malim

Editor-in-Chief

PLOS Pathogens

orcid.org/0000-0002-7699-2064

This manuscript was reviewed by three experts in Clostridioides difficile biology. Enthusiasm for the manuscript varied among these Reviewers, but even the most positive Reviewer identified some important concerns (as described in their Reviews). Some of the identified issues might be easily fixed, e.g., terminology issues for the “revertant” strains that are the focus of the study. Other important issues appear likely to require additional experimentation, e.g., concerns regarding statistical rigor/analyses/presentation and experimental design. However, addressing some critical concerns would clearly require additional experimental work, e.g., evaluating whether the suppressor mutants have other genetic or epigenetic differences with phenotypic consequences. Given the need for additional experimental work, submission of a revised manuscript would require re-review.

Reviewer's Responses to Questions

**Part I - Summary**

Reviewer #1: The article by Ormsby and colleagues makes an interesting finding. When the C. difficile FM2.5 s-layer mutant strain is inoculated into mice, suppressor strains are identified that have suppressed the original defect in the S-layer. Most of these correct the reading frame at a site that is secondary to the original mutation (and should thus be described as suppressor alleles and not revertant strains, as described throughout - RvC is a true revertant, all others are suppressors). The authors show that these suppressor strains make an in-tact S-layer and this S-layer 'sits' different in a crystal structure. The authors also find that identifying suppressor / revertant strains occurs independently of the pathogenecity locus, suggesting that the FM2.5 strain encounters conditions within the mouse that selects for S-layer positive isolates. Surprisingly, not all isolates are 'created equal' and some are more similar to the original R20291 strain while others are more similar to the FM2.5 strain. This study is likely to open the door to a number of functional studies on the C. difficile S-layer and how alterations lead to changes in permeability / physiology, thus this provides a solid level of novelty to the field.

Reviewer #2: In this report, the authors further investigate the slpA S-layer truncation mutant FM2.5 and characterize two revertants of the SlpA mutant. Using a mouse model of infection, the authors inoculated with spores of the FM2.5 mutant and observed selection for revertants within the population of cells recovered from fecal samples. These slpA revertants had varied in pathogenesis in the mouse model. The authors suggest the identification of revertants under host selection was unexpected, though prior work by this group and others firmly established the importance of the S-layer for C. difficile growth. While the revertants and their characterization are interesting, the study would benefit from more robustly powered animal experiments and subsequent analyses. Further suggestions for improvement of the manuscript are provided below.

Reviewer #3: This is a well written manuscript and describes the role of S-layer in C. difficile infection in a mouse model. Interestingly the authors isolated multiple independent revertants of an S-layer mutant. They then characterize these mutants in a mouse model of infection. They find they restore the virulence defect of the original S-layer mutant to various degrees. Overall the experiments are well performed and this is an exciting and interesting study. My biggest concern is all these experiments are based on the isolation of suppressor mutants which could have other genetic or epigenetic differences.

For example is the difference in toxin expression between RvA and RvB due solely to the different revertant or are the secondary mutations of phase variable switches driving these differences. The best way to address this is to either complement the FM2.5 mutant with RvA vs RvB to see if the changes in gene expression or rebuild the mutants using other methods.

**Part II – Major Issues: Key Experiments Required for Acceptance**

Reviewer #1: 1) The authors state levels of significance in their figure legends, but the figures themselves largely ignore significance. It is hard to tell if the values presented are significantly different, or if the authors simply forgot to label the figures properly. As an example, the FM2.5 strain has a ~1 log10 reduction in CFU in the mouse feces 24 - 72 hpi and this defect is much larger in the cecum and the colon. But if these differences are not significant (and the authors stated on line 153 that it is a trend for less recovery), then this reduces the impact of the findings.

2) Is the reason that fewer spores are recovered in Figure 4D is that what spores are formed are sensitive to heat? The authors enumerate using 56C as their selection for spores vs. vegetative cells. Given the differences in transcripts related to sporulation, it is possible that they either 1: do not make spores or 2: what spores are made cannot be enumerated.

3) Figure 5. The comparison for statistical significance should not be to the toxin control. Because strains are being compared, it is more appropriate to compare within the strains - e.g., compare to R20291, etc.

Reviewer #2: The naming of the reversion mutants RvA and RvB is generic and not indicative of the gene reversion being referenced. The authors should instead adhere to standard prokaryotic mutation nomenclature (see ASM).

The animal experiments and subsequent experimental analyses throughout the manuscript are underpowered to achieve statistically significant results. Some experiments were performed with as few as three animals and a single replicate (e.g., toxin activity, CFU). Other experiments include 2-3 fold more animals infected with the FM2.5 strain than other strains, which does not appear to be scientifically justified.

The authors delete the PaLoc (toxin locus) from the FM2.5 slpA mutant and compare it in mouse infections relative to the wildtype and FM2.5 (Fig 4). The requirement of the toxins for virulence is well established, so it is not obvious what question is being asked or answered with this experiment. Further, panel C shows the weight change for the FM2.5 without control comparisons, which lacks the context required for evaluation.

RNA-seq analysis (Fig 6) uses the FM2.5 mutant as the control comparator, rather than the wild-type strain. This comparison is not conventional and it is difficult to interpret.

Fig 2 images are not to scale, which makes it difficult to compare the sizes of the colonies and the relative difference between the mutant and control.

The abstract is long and diffuse with tangential mentions of discussion points. A more focused account of the primary findings would be helpful to the readers.

Reviewer #3: 1. My biggest concern is the potential for different the known phase variable switches in C. difficile to be in different phases in the revertants. Could this explain the colony size difference in some of the revertants? Have the authors considered reconstructing a strain with the revertant mutants to show they are responsible for the changes in virulence and gene expression? Alternatively complementing the FM2.5 mutant with RvA vs RvC to see if they recapitulate the most interesting phenotypes.

2. Sequencing experiments performed on spores vs 24, 48 etc. hpi convincingly show that the revertants are present in the spore preps. It would be helpful to compare this to the hamster experiments as well where there were no revertants recovered.

**Part III – Minor Issues: Editorial and Data Presentation Modifications**

Reviewer #1: 1) For the suppressor strains, was the whole genome re-sequenced to ensure that these are true suppressors alleles and not a contaminating strain? There are reports that mice can come pre-colonized with a contaminating C. difficile strain. I suspect these are true suppressor isolates, but for clarity, sequencing them may be useful and has become very inexpensive.

2) The importance of the RdeltaD2 S-layer needs expanded upon in the crystallographic section. It is unclear the importance of this strain and why comparisons are being made to it.

3) Figure 3 needs better labeling. Please indicate which alleles each panel represent. For example, 3A should be labeled RvB.

4) Line 450. It would be helpful to expand upon the differences between mice and hamsters for LL37, etc. and how these differences may / may not lead to suppressor phenotypes.

5) Lines 478 - 488. It is stated that >2.5% of the spores in the preparations have variations in the slpA gene. Later in the same section, the authors question why animals infected with FM2.5 do not have disease and then go on to propose reasons. One reason that is not discussed is that if the authors are giving 10^5 spores as a dose to the mice, 2.5% = 250 spores with suppressor alleles. It is likely that this low dose of 'actually infectious' C. difficile takes much longer to grow in vivo (between 8 and 9 doublings to get to the original infectious dose).

6) Line 580. Were male or female mice used?

Reviewer #2: Ln 47-49: The intent of these comparisons and the meaning of this statement is not clear.

Ln 146: This 2020 reference for the long-established correlation of weight loss and disease severity is not the first nor the most relevant reference to support the statement.

Reference to Fig 4C appears following the reference to Fig 1. Figures should be referenced in the order that they appear in the manuscript.

Fig 1 (SlpA deficient C. difficile is less pathogenic in a murine model of infection) The legend mentions statistical analyses and p values, but none are shown in the figure. Were statistical denotations left out unintentionally or were the results not significant? If not, the figure title should be amended.

Ln 207-209 mentions that the revertants RvA and RvB were the most common revertants isolated. How many revertants were identified? What other reversions occurred? Were there compensatory mutations outside of the slpA locus? How many independent clones were tested and sequenced?

Fig 2 legend (ln 229): The authors should clarify in the figure and legend that C shows truncated products for all variants.

Discussion ln 449-452: there is little known to support this conclusion. It is likely there are many factors both known and unknown that contribute to the difference in pathogenesis in the different animal models of CDI.

Fig S3: the colors representing different strains are inconsistent between the graphs.

Fig S3: Bacterial cells are resistant to killing in feces. If all cells were plated on medium that could support vegetative and spore cell growth, how do the authors know that the resulting CFU were derived from spores or vegetative cells? What controls were performed to ensure that vegetative cells were killed and only spores remained?

Reviewer #3: 1. Is it possible to run alphafold on the RvA mutant?

2. Does repeated sporulation/germination cycles in vitro produce increased recovery of revertants?

3. Can you discuss the number of times C. difficile doubles in 24hrs in mouse does that differ with a hamster?

4. Likely not in the scope of this but what is the phenotype of RvA or RvC in hamster model of infection.

PLOS authors have the option to publish the peer review history of their article (what does this mean?). If published, this will include your full peer review and any attached files.

Reviewer #1: No

Reviewer #2: No

Reviewer #3: No
---

## [Decision Letter · Decision Letter 1]

18 Apr 2023

Dear Douce,

Thank you very much for submitting your manuscript "An intact S-layer is advantageous to Clostridioides difficile within the host." for consideration at PLOS Pathogens. As with all papers reviewed by the journal, your manuscript was reviewed by members of the editorial board and by independent reviewers. In light of the reviews (below this email), we would like to invite the resubmission of a significantly-revised version that takes into account the reviewers' comments.

This revised manuscript was reviewed by three experts on Clostridioides difficile. Although Reviewer enthusiasm varied, the majority felt the manuscript was improved during revision. Nonetheless, several issues were still identified with the revised manuscript. Reviewer 1 asks that the word "revertants" be removed from Figures 2 and 4. Of more concern, Reviewer 2 has identified 4 major issues and several minor issues that need to be addressed. Some of these major issues are relatively straightforward to fix, e.g., depositing whole genome sequence data in a repository. However, other issues may involve additional experimentation. Finally, there was some Reviewer disappointment that the mutations could not be rebuilt in the parent strain, leading to some comments that the current data is being over interpreted. In general the manuscript does indicate that the findings support or suggest, rather than prove, the importance of the S-layer but this should be the uniform message in the manuscript, e.g., the title is perhaps too definitive regarding the findings.

We cannot make any decision about publication until we have seen the revised manuscript and your response to the reviewers' comments. Your revised manuscript is also likely to be sent to reviewers for further evaluation.

Sincerely,

Bruce A. McClane

Academic Editor

PLOS Pathogens

Michael Wessels

Section Editor

PLOS Pathogens

Kasturi Haldar

Editor-in-Chief

PLOS Pathogens

orcid.org/0000-0001-5065-158X

Michael Malim

Editor-in-Chief

PLOS Pathogens

orcid.org/0000-0002-7699-2064

This revised manuscript was reviewed by three experts on Clostridioides difficile. Although Reviewer enthusiasm varied, the majority felt the manuscript was improved during revision. Nonetheless, several issues were still identified with the revised manuscript. Reviewer 1 asks that the word "revertants" be removed from Figures 2 and 4. Of more concern, Reviewer 2 has identified 4 major issues and several minor issues that need to be addressed. Some of these major issues are relatively straightforward to fix, e.g., depositing whole genome sequence data in a repository. However, other issues may involve additional experimentation. Finally, there was some Reviewer disappointment that the mutations could not be rebuilt in the parent strain, leading to some comments that the current data is being over interpreted. In general the manuscript does indicate that the findings support or suggest, rather than prove, the importance of the S-layer but this should be the uniform message in the manuscript, e.g., the title is perhaps too definitive regarding the findings.

Reviewer's Responses to Questions

**Part I - Summary**

Reviewer #1: The authors have adequately addressed my prior concerns. That said, the terms 'revertants' still appear in Figure 2, 4 and should be fixed.

Reviewer #2: This report characterizes two compensatory mutants of a truncated SlpA mutant. The SlpA mutant, FM2.5, was previously shown to have defects for virulence in a hamster model of infection. The current work extends those studies to a mouse model, where they found that compensatory mutants were present in their inoculum and overtook the FM2.5 mutant during infections.

Reviewer #3: The authors have adequately addressed the most important concerns raised in the previous submission. The inclusion of the data looking at the phase variable switches provides increased confidence in the conclusions.

**Part II – Major Issues: Key Experiments Required for Acceptance**

Reviewer #1: N/A

Reviewer #2: The differences in the ratio of variants A and B in their respective input populations for animal studies is not provided (variant C ratio is provided in Fig 2g, but this variant was not shown or tested). The authors state that the ratio was “low” for the other variants, but no data are provided to suggest the starting ratios for variant A or B. Without this information, few conclusions can be drawn from the in vivo data.

The histology studies are underpowered and insufficient to draw conclusions. Fig 1e) How can PBS show high positive histology scores (appears to only have two animals)?

Full RNA-seq expression analyses should be provided. How was slpA expressed in the different strains and what fraction of the transcript was represented in the data sets? The limited data that are shown in Figure 6 appear biased towards sporulation and virulence, but there are many differentially regulated transcripts. The mutant has growth defects, which likely account for much of the differences in gene expression.

Whole-genome sequence data is not available in a repository.

Reviewer #3: (No Response)

**Part III – Minor Issues: Editorial and Data Presentation Modifications**

Reviewer #1: None

Reviewer #2: The mutations that arise in FM2.5 to restore SlpA function are compensatory, rather than revertants, since they do not revert to the original sequence.

Considering that the authors have been unable to delete the slpA gene, it seems likely that the FM2.5 mutant is a truncation with low function, rather than a true null (non-functional) mutant.

Ln 153. “..strain FM2.5 showed consistently and significantly less weight loss”. This statement is not supported by the data presented in Fig 1a. The FM2.5 mutant resulted in statistically greater weight loss at the 96 hour timepoint.

Ln 154. “measurement of total C. difficile in faecal material showed comparable levels of total numbers of bacteria shedding at 24 and 72 hpi (Fig. 1b), the recovered number of spores was significantly lower in animals infected with FM2.5 at 24 h.” The authors do not demonstrate that they measured spore viability rather than vegetative cell susceptibility to heat. Controls are needed to support this conclusion.

Ln 157. There is not sufficient data presented to support the conclusion that the mutant sporulated less efficiently in vivo.

Have the authors tested if the S-layer provides protection against the selective ChromID agar components? Because the FM2.5 mutant is highly susceptible to surface insults, the ChromID selection may bias against the FM2.5 mutant.

Ln 452-454. Have the authors considered that the differences in gene expression in the FM2.5 mutant are due to its growth defect, rather than a specific differential expression of toxin and sporulation genes?

Reviewer #3: (No Response)

PLOS authors have the option to publish the peer review history of their article (what does this mean?). If published, this will include your full peer review and any attached files.

Reviewer #1: No

Reviewer #2: No

Reviewer #3: No
---

## [Editor Report · Decision Letter 2]

14 May 2023

Dear Douce,

Thank you very much for submitting your manuscript "An intact S-layer is advantageous to Clostridioides difficile within the host." for consideration at PLOS Pathogens. As with all papers reviewed by the journal, your manuscript was reviewed by members of the editorial board and by independent reviewers. The reviewers appreciated the attention to an important topic. Based on the reviews, we are likely to accept this manuscript for publication, providing that you modify the manuscript according to the review recommendations.

A previous reviewer asked about the presence of variants A and B in the input population for the animal studies. The response to Reviewer Comments of Revision 1 indicates that the presence of variants A and B in the input were below the detection limit. Please clearly state this in the Results. The same reviewer also questioned the power of the histology analysis and the Response to Reviewer Comments indicates that, in response, more microscope sections were examined. However, line 206 of the Figure 1 legend indicates that the histologic scoring for Fig. 1e includes the caecum and upper and lower colon sections (combined) after challenge. This "lumping" is inappropriate. Please present this histopathology data separately for each tissue site rather than in combination. Should the results not be significant, they can be described as a non-significant trend. Please revise Figure 1e (and supplemental Fig 1) and the Results text accordingly. Last, you are encouraged to deposit the RNA-Seq data in a repository.

Sincerely,

Bruce A. McClane

Academic Editor

PLOS Pathogens

Michael Wessels

Section Editor

PLOS Pathogens

Kasturi Haldar

Editor-in-Chief

PLOS Pathogens

orcid.org/0000-0001-5065-158X

Michael Malim

Editor-in-Chief

PLOS Pathogens

orcid.org/0000-0002-7699-2064

A previous reviewer asked about the presence of variants A and B in the input population for the animal studies. The response to Reviewer Comments of Revision 1 indicates that the presence of variants A and B in the input were below the detection limit. Please clearly state this in the Results. The same reviewer also questioned the power of the histology analysis and the Response to Reviewer Comments indicates that, in response, more microscope sections were examined. However, line 206 of the Figure 1 legend indicates that the histologic scoring for Fig. 1e includes the caecum and upper and lower colon sections (combined) after challenge. This "lumping" is inappropriate. Please present this histopathology data separately for each tissue site rather than in combination. Should the results not be significant, they can be described as a non-significant trend. Please revise Figure 1e (and supplemental Fig 1) and the Results text accordingly. Last, you are encouraged to deposit the RNA-Seq data in a repository.

Reviewer Comments (if any, and for reference):

Figure Files:

Data Requirements:

Reproducibility:

References:

---

## [Editor Report · Decision Letter 3]

31 May 2023

Dear Douce,

We are pleased to inform you that your manuscript 'An intact S-layer is advantageous to Clostridioides difficile within the host.' has been provisionally accepted for publication in PLOS Pathogens.

Best regards,

Bruce A. McClane

Academic Editor

PLOS Pathogens

Michael Wessels

Section Editor

PLOS Pathogens

Kasturi Haldar

Editor-in-Chief

PLOS Pathogens

orcid.org/0000-0001-5065-158X

Michael Malim

Editor-in-Chief

PLOS Pathogens

orcid.org/0000-0002-7699-2064

The authors have made the requested modifications so the manuscript is now acceptable.
---

## [Editor Report · Acceptance letter]

15 Jun 2023

Dear Douce,

We are delighted to inform you that your manuscript, "An intact S-layer is advantageous to Clostridioides difficile within the host.," has been formally accepted for publication in PLOS Pathogens.

Best regards,

Kasturi Haldar

Editor-in-Chief

PLOS Pathogens

orcid.org/0000-0001-5065-158X

Michael Malim

Editor-in-Chief

PLOS Pathogens

orcid.org/0000-0002-7699-2064